# Dynamics of sociopolitical polarization and effects of misperception-correcting information around the 2022 Brazilian elections

Anna Petherick [1] ✉, Guilherme A. Ramos [2,3], Rodrigo Furst [4],
Rafael Goldszmidt[5] & Eduardo B. Andrade [6]

Affective polarization is little-studied in systems that present ambiguity in relevant political groups. We assess dynamic patterns in intergroup liking and perceived intelligence over 5 waves of a pre-registered survey with 4 panel waves (N ≃ 2,000-3,000) conducted before, during, and after the 2022 Brazilian elections. Given the nature of Brazil's political system, we document horizontal affective polarization of combinations of political ingroups and outgroups. These include left-right self-placed position-holders, negative partisans, and supporters of the most important presidential candidates, "Lula" and "Bolsonaro". Categorized by first-round vote intention, the latter grouping (lulistas vs. bolsonaristas) shows particularly intense affective polarization before the election campaign, which remains stable through the elections, thereafter fading approximately symmetrically. We also demonstrate that experimental treatment conditions correcting misperceptions of group stances on single, controversial policies reduce affective polarization by up to 6.66 percentage points (liking thermometer), and, in some instances, also shift policy support.

Affective polarization among ordinary citizens has been linked to many normatively negative political, social and economic outcomes[1-3]. Not only do people who exhibit dislike and distrust towards those in different socio-political groups tend to mis-imagine the beliefs of the other[4], they also socially distance themselves, avoiding ties such as romantic partnerships[5] and living in the same neighborhood[6] as their outgroup(s), creating echo chambers of opinion. Individuals registering high levels of affective polarization have been shown to exhibit bias in merit-based decisions, including in the dispensation of scholarships[7] and the acceptance of employment opportunities and pay[8]. These various tendencies, en masse, have the potential to drive and sustain profound divisions in many societies around the world, promote the

decay of democratic norms[9] and even increase support for political violence[10]. Given such serious prospects, understanding the dynamics of affective polarization and identifying phenomena and light-touch interventions that can soften it in challenging contexts are of critical importance.

Most research into how to best measure and reduce affective polarization has focused on the exceptional case of the United States, where political ingroups and outgroups are relatively easily defined[11,12], and aggravators such as 'sorting' have been identified that are not in all cases plausibly prominent causes elsewhere[13]. A growing literature suggests that affective polarization is a common phenomenon in countries in Europe[14], following recent developments in the

[1]Blavatnik School of Government, University of Oxford, Oxford, UK. [2]Rochester Institute of Technology, Saunders College of Business, New York, NY, USA. [3]Pensi Institute, São Paulo, Brazil. [4]King's Business School, King's College London, London, UK. [5]Brazilian School of Public and Business Administration, Getulio Vargas Foundation, Rio de Janeiro, Brazil. [6]Business School, Imperial College London, London, UK. ✉e-mail: anna.petherick@bsg.ox.ac.uk

assessments of affective polarization in multiparty systems[15,16]. Attempts to measure the phenomenon elsewhere in the world, including in Asia[17], Africa[18] and Latin America[19] are fewer. There is also a paucity of research exploring measurement options in political systems that present significant ambiguity in the relevant types of political ingroups and outgroups[20,21]. And while various misperception-correction treatments to reduce affective polarization have been tested sufficiently frequently in the United States to warrant meta-assessment in two recent papers[22,23], they have been far less commonly trialled in other countries, and, in our view, their varied effects warrant further reflection.

This paper contributes to filling these gaps by exploring the case of Brazil, a country that poses significant challenges for the measurement of affective polarization, and, according to recent scholarship, also for its attenuation[24]. Brazil is a tricky case for affective polarization measurement in part because it is one of the most fragmented political systems in the world[25] where electorally relevant political parties display wide variation in ideological consistency[26], in what Scott Mainwaring referred to as an inchoate party system[27]. The problem is not merely party fragmentation, but politicians' sometimes-rampant party switching[28] (muddling the symbolic tie between a political leader and party represented), on top of limited (and uneven) party identification among citizens[29], who generally experience a political environment that is largely person- (or politician-) rather than party-centered, and in which negative partisanship (in both anti-party[30] and anti-politician varieties) is rife.

Drawing on data from five nationally representative survey waves, with panel responses composing the final four, we employ a diverse set of ingroup-outgroup assessments to follow dynamic variations in affective polarization among groups of citizens throughout the 2022 electoral period, and into 2023. The electoral period that we follow was a fraught one. After several months of campaigning, Jair Bolsonaro's presidential re-election bid fell to competition from former president, Luiz Inácio Lula da Silva, in first-round and then run-off elections in October 2022. Lula took office on 1 January 2023, and a week later rioters stormed key government buildings in the capital, in events echoing those of 6 January 2021 in the U.S.

We also assess the effectiveness of four different, single-issue experimental treatments to correct misperceptions of ingroup and outgroup stances towards controversial public policies. The experimental treatment conditions that we select to correct misperceptions deliberately pertain to controversial issues, in the sense that they are morally charged topics about which opposing opinions are commonplace, and they feature with some regularity in public discussion. While views on abortion in the first three months of pregnancy, for example, may be entrenched, the moral character of this issue also implies that misperception corrections may be highly informative[31], and the topic's frequent featuring in public debate suggests the possibility of increased accessibility[32,33].

Our pre-registered hypotheses anticipated that: i) the moment of the elections would heighten affective polarization, and ii) experimental treatments correcting misperceptions of group positions on controversial issues would reduce it. Our results show large and relatively symmetrical affective polarization, for both Lula supporters (hereafter, lulistas) and Bolsonaro supporters (hereafter, bolsonaristas), from before the official campaign period to the elections. We then find a modest reduction in affective polarization post-election for these groups, driven mostly by improvements in outgroup affect—a dynamic pattern that was less apparent for other political group definitions (e.g., by self-placement on a left-right ideology scale). While the impacts of our experimental treatments are modest with respect to some different approaches to reducing affective polarization, in comparison to other policy-stance misperception corrections, the effect sizes of our most morally charged interventions are substantial.

## Results
### Dynamics of affective polarization throughout the 2022 elections

We operationalized affective polarization as the difference between ingroup and outgroup evaluations using feeling thermometer (like-dislike) scales across five survey waves. These spanned periods before the elections (waves 1 and 2, collected in April 2022 and July 2022, respectively), between the first and run-off elections (wave 3, in October 2022), and after the elections (waves 4 and 5, which started in November 2022 and late-January 2023, respectively). We measured perceived intelligence as well as liking, as the former is regarded as a coarser instrument to capture more extreme sentiments that stray towards outgroup dehumanization, and may also capture an aspect of distrust in the form of perceived competence. Respondents were asked to score sentiment towards groups of citizens of various political persuasions (i.e., "horizontal"[34] or "social"[35] polarization), providing assessments of different camps of distinct definitional types of political groups: left-right position-holders (defined by respondent self-placement on a left-right ideology scale), lulistas and bolsonaristas (defined by intention-to-vote for Lula or Bolsonaro in the first-round presidential election), anti-lulistas and anti-bolsonaristas (those who stated they would never vote for these respective candidates), and "petistas" and "anti-petistas" (supporters and those who stated they would never vote for the Workers' Party, Brazil's only party to enjoy significant levels of voter identification[29]). The first survey wave enabled us to compare frequencies of respondents' ingroup self-categorizations, as well as the frequencies of co-occurrences of different self-categorizations (see Supplementary Fig. 1 for details of co-occurrences, including two varieties of pro-politician categorization: "being comfortable calling oneself a lulista/bolsonarista" in addition to intention-to-vote for Lula or Bolsonaro in the first-round elections, the categorization used throughout the Results).

Beyond the question of which groups to measure, the Brazilian context presents additional complexities. For example, recent developments in the assessment of affective polarization in multiparty systems weight liking scores using party vote share[15,16,35]. These are not straightforwardly applicable in the Brazilian case since the huge number of electorally relevant parties means that thorough liking batteries would be impractically long. Politicians' frequent party switching also makes this a problematic approach to weight their (non-) supporter groups, as per Reiljan et al.[35], because it implies weak (and variable) symbolic correspondence between politician and party (although other measures of electoral relevance could be used). Indeed, Bolsonaro was not a member of any party for a portion of his presidency and has represented nine different parties during his career. Another issue is the non-inclusion in aggregate measures of affective polarization of respondents who do not identify with an ingroup, and thus inevitably have no individual ingroup-outgroup feeling-thermometer gap. Some scholars minimize this problem using definitions such as leaning towards one political party over others (i.e., survey respondents feeling "not close" but "a little closer to")[15]; others instead do not require respondents to express even weak support, and define ingroup as the most-liked target (e.g., party), among multiple assessed targets of the same type (i.e., other parties)[16]. For simplicity and consistency across political group types, we considered single (rather than composite) outgroups, and did not apply weighting. Hence, pro-politician outgroups were considered to be supporters of the one other important presidential candidate (i.e., lulistas vs. bolsonaristas), although alternative approaches are possible using our data (i.e., lulistas vs. anti-lulistas). Using intention-to-vote for Lula or Bolsonaro in first-round elections to define lulista and bolsonarista groupness excluded 36.82% of respondents in the first wave. This is a similar percentage to those Reiljan[15] reports for his affective polarization index applied to European countries. Using ideological self-placement (left-wing and right-wing) to define groups excluded 43.87%

of respondents, and using our anti-politician definition excluded 22.38%. Of note, given recent arguments about the depth and persistence of the petistas vs. anti-petistas division in Brazilian politics[24], our first-wave feeling thermometer assessment excluded 51.13% of respondents when the petistas vs. anti-petistas grouping was used, and had lower affective polarization scores (liking: M = 46.87, intelligence: M = 41.57) than lulistas vs. bolsonaristas (liking: M = 59.20, intelligence: M = 53.85). For these reasons, we pre-registered experimental treatments to reduce affective polarization among lulistas and bolsonaristas across waves 2 to 5.

We also obtained data in the first wave about policy opinions on controversial issues that we then used to construct our experimental treatments (see Supplementary Table 1 for details). The second wave recruited a different sample of participants, who were then repeatedly contacted over the subsequent waves, creating a panel structure (see details of sampling in the Methods section and descriptive analysis of the sample over waves in Supplementary Table 2). First-wave data were collected before several preregistrations; however, as explained in preregistration protocols, hypotheses to be tested pertained to phenomena that were yet to occur (e.g., the evolution of polarization over time, and effects of misperception-correction treatments).

We used linear regression models with standard errors clustered by respondent to test our preregistered hypotheses. Unless otherwise noted, results reported follow pre-registered analysis plans. We also conducted exploratory Bayesian analyses to provide complementary insights. In this section, we present classical two-sided test statistics and p-values for all results. To conserve space in the main manuscript, we include Bayes factors for every null classical test result as well as for results with Bayes factors pointing toward the null rather than the alternative ($BF_{10} < 1$) for which classical test results are significant ($p < 0.05$). Bayes factors pertaining to other significant findings and to equivalence tests are reported in the Supplementary Materials. We opted for linear models rather than models for bounded variables as they allow for easier interpretation of coefficients, and because the dependent variables were not skewed (liking: skewness = 0.038; perceived intelligence: skewness = 0.008). For hypotheses related to changes in polarization over time, dependent variables were liking/perceived intelligence. Predictors included dummy variables representing mid-election and post-election periods, and a dummy variable indicating if the evaluation refers to an ingroup or outgroup target; their interactions capture changes in polarization over time. As controls, we included demographic characteristics of the respondent and treatment conditions. The change in the cohort of participants between wave 1 and the panel of waves 2 to 5 precludes the use of participant fixed effects to assess dynamic changes in polarization (see detailed model specifications in Methods section, and Supplementary Table 3 for participants' exposure to treatments across waves).

While our pre-registration specified models using either linear time trends or individual wave dummies, we report results grouping waves by election phase (before, during, after). This approach strikes a balance between parsimony and flexibility, allowing us to capture distinct periods more effectively than a linear trend. Figure 1 presents the evolution of affective polarization before, during, and after the elections using two of our ingroup-outgroup definitions: the pro-politician (lulistas and bolsonaristas) and ideological identification (left-wing and right-wing) (see Supplementary Fig. 2 and Supplementary Table 4 for pre-registered results disaggregated by wave and considering time trends, and Supplementary Fig. 3 for results using anti-politician groupings (those who would never vote for Lula/Bolsonaro). To assess if these results were affected by differential attrition, whereby the probability of dropping out of the panel is associated with individual level of affective polarization, we performed sensitivity analyses with alternative imputations of the measures for non-respondents; results remained largely unchanged (see Supplementary Fig. 4). Moreover, results for trends considering only respondents who

were randomly assigned to the control group in our survey experiments are also consistent (as opposed to using the whole sample controlling for effects of the misperception-correcting information treatment) (see Supplementary Fig. 5). Results also remain largely unchanged using mixed-effects regression (Supplementary Fig. 6).

Inspecting the dynamics of intergroup sentiment between lulistas and bolsonaristas, we can visually observe in Fig. 1 high levels of affective polarization in the pre-election period (i.e., waves 1 and 2), with large differences between average liking of ingroups (M = 71.88, SD = 27.04) and outgroups (M = 17.16, SD = 25.54), as well as between perceived intelligence of ingroups (M = 70.13, SD = 25.95) and outgroups (M = 19.86, SD = 25.63). Although this observed polarization was roughly symmetrical, nuances are noticeable (Supplementary Figs. 7–9). Our exploratory analyses (see Supplementary Table 4 for pre-registered results) found no evidence of changes in affective polarization during the election wave compared to the pre-election period, both in terms of liking (b = −1.19, t(2,581) = −1.11, p = 0.266, 95% CI [−3.28, 0.91], standardized coefficient = −0.01) and perceived intelligence (b = −0.35, t(2,581) = −0.35, p = 0.728, 95% CI [−2.30, 1.61], standardized coefficient = −0.00). To explore if these results imply evidence of the absence of effects, we calculated Bayes factors in the direction of the alternative hypothesis ($BF_{10}$) to quantify how likely the alternative hypothesis was compared to the null. Bayes factors were 0.05 for liking and 0.03 for perceived intelligence, which suggests that the observed data were, respectively, 20 and 33 times more likely under the null hypothesis than under the alternative hypothesis (see Supplementary Tables 5a-5m for a summary of Bayes factors and equivalence tests for all results, and Methods section for details on the calculation procedures).

We observed, however, a modest reduction post-election (average of waves 4 and 5 vs. average of waves 1 and 2; liking: b = −4.46, t(2,581) = −4.52, p < 0.001, 95% CI [−6.39, −2.52], standardized coefficient = −0.05; perceived intelligence: b = −4.17, t(2,581) = −4.31, p < 0.001, 95% CI [−6.07, −2.27], standardized coefficient = −0.05). This reduced affective polarization was driven by an increase in outgroup evaluation (liking: b = 7.12, t(2,581) = 9.45, p < 0.001, 95% CI [5.64, 8.60], standardized coefficient = 0.09; perceived intelligence: b = 6.61, t(2,581) = 8.83, p < 0.001, 95% CI [5.14, 8.08], standardized coefficient = 0.09) along with a contemporaneous, smaller increase in ingroup evaluation that was not supported by Bayes factors (liking: b = 2.66, t(2,581) = 3.50, p < 0.001, 95% CI [1.17, 4.16], standardized coefficient = 0.03, $BF_{10}$ = 0.04; perceived intelligence: b = 2.44, t(2,581) = 3.29, p = 0.001, 95% CI [0.99, 3.89], standardized coefficient = 0.03, $BF_{10}$ = 0.06). Levels of affective polarization vary considerably when using alternative group definitions, as the lower panel in Fig. 1 illustrates. Trends also vary, though in all cases, changes in affective polarization over time are of small magnitude (see Supplementary Figs. 3 and 10 for the dynamics of anti-politician sentiment and left-right ideological identification considering target groups that talk a little vs. a lot about politics). We found no meaningful changes during the period of analysis for meta-polarization using the intention-to-vote pro-politician definition, except for a small increase when comparing perceived intelligence scores during (vs. before) the elections (b = 1.79, t(2,923) = 2.17, p = 0.030, 95% CI [0.17, 3.40], standardized coefficient = 0.02) based on classical hypothesis testing, but with some evidence for the null considering Bayes factors ($BF_{10}$ = 0.10); see Supplementary Fig. 11 for detailed results and Supplementary Tables 5a-5m for Bayes factors and equivalence tests for all findings). Policy opinions showed little change over our period of analysis (Supplementary Fig. 12).

Our fourth and fifth waves presented the opportunity to examine the potential influence of contextual factors other than elections on levels of affective polarization, as the fourth wave was fielded throughout the 2022 FIFA World Cup, and the fifth approximately three weeks after 8 January 2023, when rioters supporting Bolsonaro

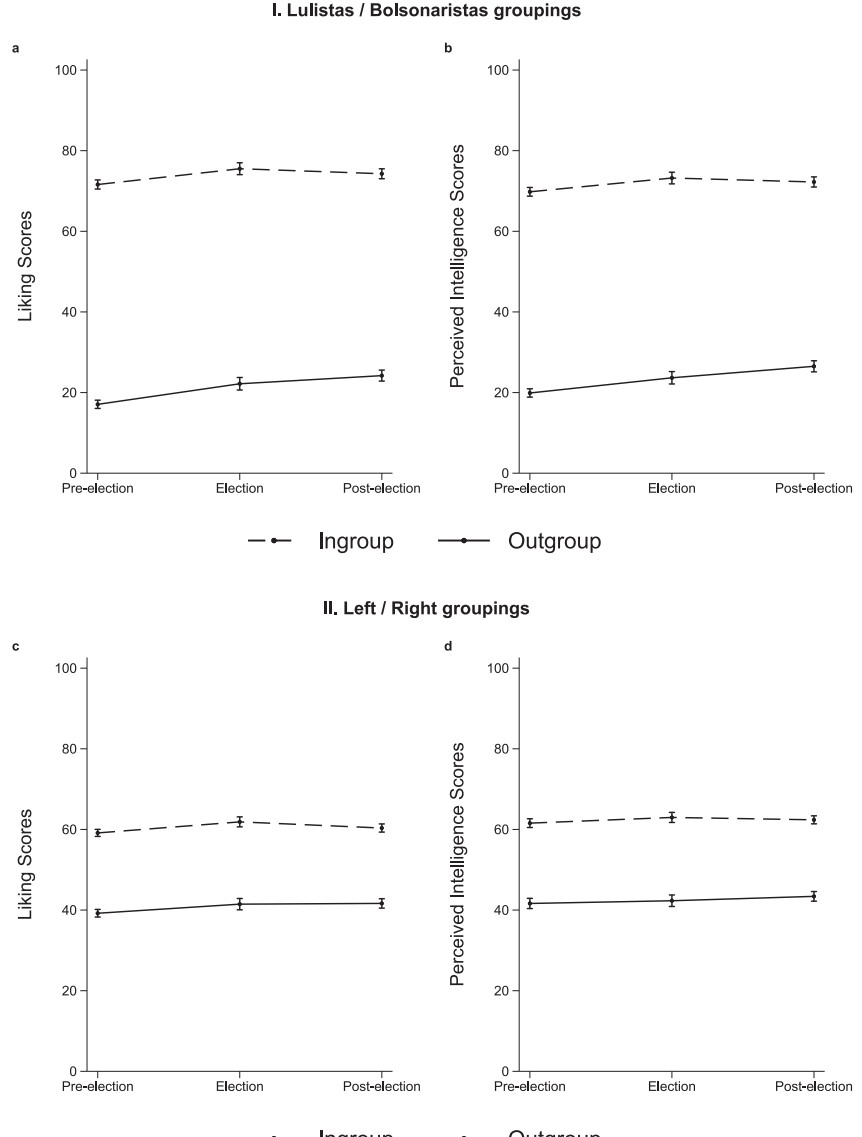

**Fig. 1 | Evolution of affective polarization.** Panels **a** ($N = 13{,}594$) and **c** ($N = 11{,}244$) depict how average liking scores fluctuate before, during, and after the election period for political ingroups (individuals who share the respondent's political self-categorization) and outgroups (individuals who hold a different political self-categorization). Panels **b** ($N = 13{,}594$) and **d** ($N = 9{,}332$) depict equivalent effects for perceived intelligence. These effects were estimated using linear regression models, controlling for participants' age, self-reported sex, household income, educational attainment, region of residence, and dummies for wave-specific treatment conditions. This figure uses the grouping criteria: (i) reported intentions to vote for Lula (lulistas) or Bolsonaro (bolsonaristas) in the first election round (Panel I), and (ii) ideological left-right self-identification (Panel II). To mitigate endogeneity concerns (i.e., participants changing both their political identification and affective polarization scores over time due to unobserved factors), we used as grouping criteria the participants' voting intentions and ideological identification reported in the first wave they participated in. Panel I excludes participants who did not report intentions to vote for either Lula or Bolsonaro and Panel II excludes participants who reported a center ideology, or who chose not to identify as left, center, or right, or who lacked knowledge about the meaning of these ideological labels. Data are presented as model-predicted mean values, with error bars indicating 95% confidence intervals. Models were estimated with clustered standard errors at the individual level. All statistical tests were two-sided.

invaded the National Congress, the Supreme Court, and the Presidential Palace. Exogenous events that raise the salience of national identities, such as the Olympic Games, have been found to reduce affective polarization by lowering the salience of intra-national identities, such as political groups, relative to a national identity[36]. Protests, meanwhile, that pit one political group against another have been found to increase affective polarization[37]. Hence, to examine the impact of the World Cup, the fourth wave was divided into eight sub-waves, with the first fielded just prior to Brazil's first match of the competition, and the remaining sub-waves after each of Brazil's subsequent matches (and after the matches that Brazil would have played had the team not been eliminated from the competition in the quarter finals). Overall, noting the inevitably limited statistical power of these sub-samples, our preregistered analyses—and subsequent exploratory Bayes Factors—found evidence that affective polarization was not associated with the Brazilian team's performance (i.e., wins and losses) (after victories: liking: $BF_{10} = 0.05$; perceived intelligence: $BF_{10} = 0.05$; and after losses: liking: $BF_{10} = 0.07$; perceived intelligence: $BF_{10} = 0.11$). However, in an exploratory analysis, we found limited evidence for separate trends across sub-waves for respondents who considered the Brazilian team's football shirt to be a partisan symbol ($N = 622$), instead of a national one ($N = 1{,}266$) (see Supplementary Fig. 13). A preregistered model examining wave-specific heterogeneity allowed us to compare wave 4 (the average of these sub-waves) to wave 3 (the period

between the first-round election and the run-off), revealing a small reduction in affective polarization, but with Bayes factors indicating some evidence for the null (liking: b = −2.95, $t(2,581)$ = −2.39, $p$ = 0.017, 95% CI [−5.38, −0.53], standardized coefficient = −0.02, $BF_{10}$ = 0.18; perceived intelligence: b = −2.99, $t(2,581)$ = −2.59, $p$ = 0.010, 95% CI [−5.25, −0.72], standardized coefficient = −0.03, $BF_{10}$ = 0.25). Note that this wave of the post-election period is perfectly confounded with the World Cup's timing. From wave 4 to wave 5, we found no evidence of changes in affective polarization in liking (b = −0.61, $t(2,581)$ = −0.52, $p$ = 0.604, 95% CI [−2.92, 1.70], standardized coefficient = −0.01) or perceived intelligence (b = −1.62, $t(2,581)$ = −1.46, $p$ = 0.146, 95% CI [−3.80, 0.56], standardized coefficient = −0.01), as confirmed by Bayes factors (liking: $BF_{10}$ = 0.02; perceived intelligence: $BF_{10}$ = 0.04). For further analyses on polarization during the World Cup and after the 8 January riots, see Supplementary Note 1 and Supplementary Figs. 14–16 in Supplementary Materials.

## Treatment effects of misperception-correcting information on affective polarization

Affective polarization is widely associated with differential accuracy (estimating ingroup positions more accurately than outgroup ones[38]), and the exaggeration of differences between one's ingroup and outgroup[39]. Thus, we examined whether reducing misperceptions of ingroup and outgroup support for policies—relating to the controversial issues of abortion, Amazonian deforestation and affirmative action—could attenuate it.

Starting with wave 2, we randomly assigned respondents to control (not receiving misperception-correcting information before evaluating liking and intelligence of groups) or to receiving one of two treatment conditions of misperception-correcting information prior to encountering feeling thermometer questions. Our treatment conditions first provided respondents with a neutral briefing about a public policy, stating that some people support it while others do not (see Supplementary Table 1 for text of treatment conditions). Respondents assigned to a treatment condition were then asked to estimate how many lulistas and bolsonaristas out of 10, on average, support that policy. They were subsequently informed about those groups' actual level of support, using our data on policy support from wave 1, to reveal the gap between their own estimations and the earlier findings of the same survey that they were taking part in.

In each wave, we assessed the effects of misperception-correcting information relating to two different controversial policies. In waves 2 and 3, we used as treatment conditions: (i) a policy of legalization of abortion up to the third month of pregnancy, and (ii) a policy of zero deforestation in the Amazon. Respondents were initially randomly assigned to control or to either treatment condition, with those assigned to control in wave 2 also assigned to control in wave 3. Those assigned to a treatment condition in wave 2 received the alternative condition in wave 3. In waves 4 and 5, we assessed treatment conditions about affirmative action policies that assist those from low-income households, and Afro-Brazilians and indigenous people in gaining access to higher education[40]: (iii) social quotas, and (iv) racial quotas, respectively. The same approach to assignment was repeated as per waves 2 and 3. In wave 4, respondents were randomized independently of their assignments in prior waves. Then those assigned to control remained there across waves 4 and 5, and those assigned to a treatment condition in wave 4 were counterbalanced in wave 5. Supplementary Table 3 summarizes the flow of all panel respondents through assignment to control or experimental conditions for all survey waves.

The information about Lulista and Bolsonarista average levels of actual support for each of these controversial policies, as recorded in wave 1, diverged significantly from respondents' ex-ante perceptions (see Supplementary Table 6). For example, while lulistas believed, on average, that 59.78% of their ingroup supported the legalization of abortion in the first 3 months of pregnancy, bolsonaristas, on average, held much larger misconceptions, estimating the support of lulistas (their outgroup) at 81.48%. Our results indicate an actual figure of 46.37%, below both of those numbers.

To test randomized treatment effects about affective polarization, our preregistered analyses used data from waves 2–5, employing linear regression models and respondent fixed effects rather than demographic controls to reduce error variance unrelated to the treatment, which is possible given participants were repeated across these waves. Again, dependent variables were liking/perceived intelligence. The independent variables included a dummy indicating if the target of the evaluation is an ingroup (vs outgroup, varying within subjects), and its interaction with dummy variables representing treatment conditions (varying between subjects). The coefficient of this interaction term captures treatment effects (i.e., difference in polarization across treatment conditions). We consider only effects of the treatment in the same wave it was dispensed, as our design is not suited to the evaluation of long-term effects. To assess our main treatment effects, we combine data from waves with identical treatment conditions to estimate the average effects of each condition (for example, the assessment of the Amazon condition combines data from waves 2 and 3). Results remain largely unchanged using mixed-effects models with random intercepts and trends, as recommended by Curran and Bauer[41] (Supplementary Fig. 17).

Figure 2 shows treatment effects for liking and perceived intelligence for each treatment condition, consolidating identical treatments over consecutive waves (see Supplementary Figs. 18–19 for results by wave, and Supplementary Table 7 for pre-registered regression results). The results confirm our preregistered expectations: offering information to correct participants' misperceptions tends to reduce affective polarization (left-hand columns), and this effect is mainly driven by less negative assessments of outgroups (right-hand columns). For instance, upon learning the percentage of lulistas and bolsonaristas who support the legalization of abortion in the first 3 months of pregnancy (a minority of both groups), both lulistas and bolsonaristas perceived their outgroup as more likable (b = 6.64, $t(997)$ = 4.06, $p$ < 0.001, 95% CI [3.43, 9.85], standardized coefficient = 0.03, for lulistas; b = 5.41, $t(841)$ = 3.38, $p$ = 0.001, 95% CI [2.27, 8.55], standardized coefficient = 0.02, for bolsonaristas) and intelligent (b = 7.63, $t(997)$ = 4.74, $p$ < 0.001, 95% CI [4.47, 10.78], standardized coefficient = 0.05, for lulistas; b = 8.45, $t(841)$ = 5.33, $p$ < 0.001, 95% CI [5.34, 11.56], standardized coefficient = 0.04, for bolsonaristas). Overall, outgroup thermometer scores warm significantly in response to all four treatment conditions (Fig. 2, first bar, third columns), for both liking (abortion: b = 6.07, $t(1,839)$ = 5.28, $p$ < 0.001, 95% CI [3.82, 8.33], standardized coefficient = 0.03; Amazon: b = 3.62, $t(1,839)$ = 3.33, $p$ = 0.001, 95% CI [1.49, 5.76], standardized coefficient = 0.01; social quotas: b = 3.34, $t(1,437)$ = 2.36, $p$ = 0.019, 95% CI [0.56, 6.12], standardized coefficient = 0.02; racial quotas: b = 4.17, $t(1,437)$ = 2.87, $p$ = 0.004, 95% CI [1.32, 7.02], standardized coefficient = 0.02) and perceived intelligence (abortion: b = 8.00, $t(1,839)$ = 7.07, $p$ < 0.001, 95% CI [5.78, 10.22], standardized coefficient = 0.04; Amazon: b = 6.79, $t(1,839)$ = 6.19, $p$ < 0.001, 95% CI [4.64, 8.94], standardized coefficient = 0.03; social quotas: b = 3.81, $t(1,437)$ = 2.67, $p$ = 0.008, 95% CI [1.01, 6.61], standardized coefficient = 0.02; racial quotas: b = 5.22, $t(1,437)$ = 3.63, $p$ < 0.001, 95% CI [2.40, 8.05], standardized coefficient = 0.03). For the perceived intelligence measure, the changes in outgroup affect were sufficient to drive significant reductions in affective polarization for all treatments.

Specifically, we observed significant treatment effects leading to a reduction in affective polarization for the abortion condition (liking: b = −6.66, $t(1,839)$ = −3.94, $p$ < 0.001, 95% CI [−9.97, −3.35],

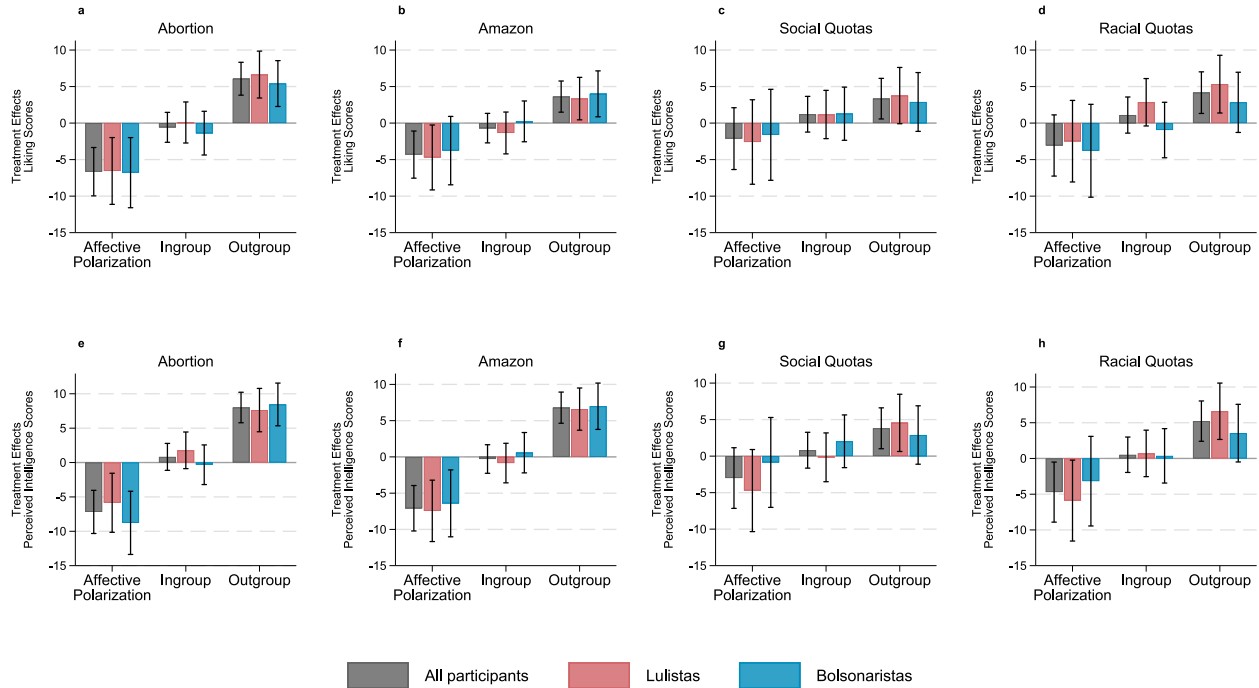

**Fig. 2 | The effect of providing misperception-correcting information on affective polarization.** Panel **a** (N = 6,530) depicts the average effect of misperception-correcting information about support for legalizing abortion in the first 3 months of pregnancy on liking scores for political ingroups (individuals who share the respondent's political self-categorization), outgroups (individuals who hold a different political self-categorization), and their difference (affective polarization). Panels **b** (N = 6,530), **c** (N = 4,912), and **d** (N = 4,912) depict equivalent effects of misperception-correcting information about support for zero deforestation in the Amazon, access to universities through social and racial quotas, respectively. Panels **e**–**h** replicate these effects for the perceived intelligence scores. This figure considers as grouping criterion the reported intentions to vote for Lula (lulistas) or Bolsonaro (bolsonaristas) in the first election round. These effects were estimated using linear regression models with fixed-effects and clustered standard errors at the individual level. Data are presented as estimated mean changes, with error bars indicating 95% confidence intervals. All statistical tests were two-sided.

standardized coefficient = −0.03, for all participants; b = −6.56, t(997) = −2.81, p = 0.005, 95% CI [−11.13, −1.98], standardized coefficient = −0.03, for lulistas; b = −6.79, t(841) = −2.77, p = 0.006, 95% CI [−11.59, −1.99], standardized coefficient = −0.03, for bolsonaristas; perception of intelligence: b = −7.18, t(1,839) = −4.48, p < 0.001, 95% CI [−10.32, −4.04], standardized coefficient = −0.04, for all participants; b = −5.85, t(997) = −2.67, p = 0.008, 95% CI [−10.14, −1.56], standardized coefficient = −0.03, for lulistas; b = −8.77, t(841) = −3.74, p < 0.001, 95% CI [−13.37, −4.17], standardized coefficient = −0.04, for bolsonaristas). We also observed significant treatment effects leading to a reduction in affective polarization for the Amazon condition (for all participants, liking: b = −4.32, t(1,839) = −2.62, p = 0.009, 95% CI [−7.54, −1.09], standardized coefficient = −0.02; perception of intelligence: b = −7.08, t(1,839) = −4.43, p < 0.001, 95% CI [−10.21, −3.95], standardized coefficient = −0.04). Within this, bolsonaristas showed significant effects for the Amazon condition for intelligence but not liking, with some evidence of the null hypothesis from Bayes factors for liking (liking: b = −3.77, t(841) = −1.58, p = 0.115, 95% CI [−8.45, 0.92], standardized coefficient = −0.02, BF₁₀ = 0.16; perception of intelligence: b = −6.41, t(841) = −2.72, p = 0.007, 95% CI [−11.02, −1.79], standardized coefficient = −0.03). Lulistas, on the other hand, showed significant effects for both liking and intelligence, but with some evidence for the null with respect to liking according to Bayes factors (liking: b = −4.71, t(997) = −2.08, p = 0.038, 95% CI [−9.15, −0.26], standardized coefficient = −0.02, BF₁₀ = 0.39; perception of intelligence: b = −7.44, t(997) = −3.44, p < 0.001, 95% CI [−11.68, −3.21] standardized coefficient = −0.04).

Among the two affirmative action conditions, treatment effects on affective polarization were significant only for the racial quotas treatment according to the perceived intelligence measure, though with some evidence supporting the null according to Bayes factors (for all participants: b = −4.71, t(1,437) = −2.20, p = 0.028, 95% CI [−8.91, −0.51], standardized coefficient = −0.02, BF₁₀ = 0.80). While similar results were obtained for the abortion and Amazon treatments when considering the alternative political group definitions, the effects for affirmative action policies were even less apparent (Supplementary Fig. 20 for right- vs. left-wing, Supplementary Fig. 21 for anti-politician groups, and Supplementary Tables 5a-5m for Bayes factors). Exploratory analyses found no evidence that the treatment effects of the abortion or Amazon conditions were different across waves 2 and 3 (ps > 0.327), nor that the treatment effects for the social quotas condition were different across waves 4 and 5 (ps > 0.850). These null results were supported by Bayes factors for the abortion (liking: BF₁₀ = 0.06; perceived intelligence: BF₁₀ = 0.07), Amazon (liking: BF₁₀ = 0.04; perceived intelligence: BF₁₀ = 0.08), and social quotas conditions (liking: BF₁₀ = 0.05; perceived intelligence: BF₁₀ = 0.05). There was a significant difference, however, in the treatment effects of the racial quotas condition across waves 4 and 5 (liking: b = 10.38, t(1,437) = 2.86, p = 0.004, 95% CI [3.26, 17.50], standardized coefficient = 0.02; perceived intelligence: b = 8.26, t(1,437) = 2.30, p = 0.021, 95% CI [1.23, 15.29], standardized coefficient = 0.05), with a significant effect only in wave 4 (liking: b = −8.27, t(1,179) = −2.98, p = 0.003, 95% CI [−13.71, −2.84], standardized coefficient = −0.03; perceived intelligence: b = −8.87, t(1,178) = −3.28, p = 0.001, 95% CI [−14.17, −3.57], standardized coefficient = −0.04), and not in wave 5 (liking: b = 2.11, t(1,275) = 0.74, p = 0.457, 95% CI [−3.45, 7.67], standardized coefficient = 0.01; perceived intelligence: b = −0.61, t(1,276) = −0.21, p = 0.832, 95% CI [−6.26, 5.04], standardized coefficient = −0.003). The null effects in wave 5 were supported by Bayes factors (liking: BF₁₀ = 0.07; perceived intelligence: BF₁₀ = 0.05).

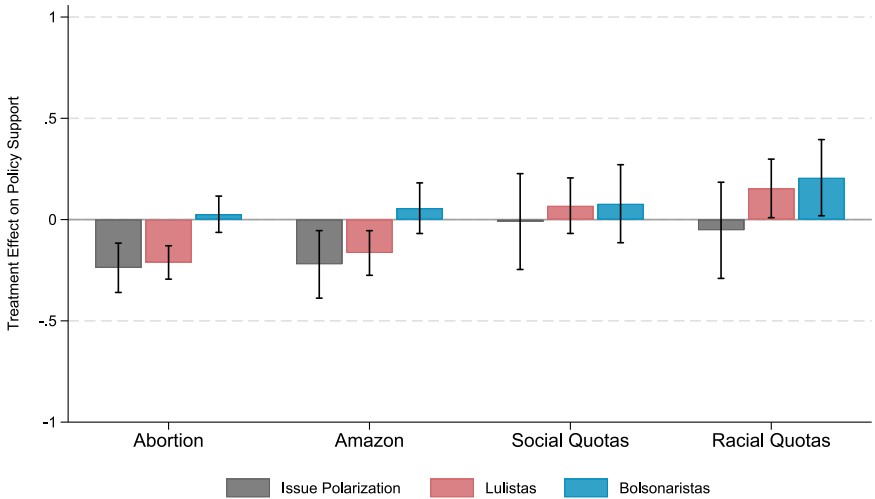

**Fig. 3 | The effect of providing misperception-correcting information on policy support.** This figure depicts changes in participants' policy support towards abortion within the first three months of pregnancy ($N = 4,733$), zero deforestation ($N = 4,744$), social quotas ($N = 3,030$), and racial quotas ($N = 3,011$) among respondents who reported intentions to vote for Lula (lulistas) or Bolsonaro (bolsonaristas) in the first election round, and their difference in support (issue polarization), after versus before receiving misperception-correcting information about the respective policy. These effects were estimated using linear regression models controlling for age, self-reported sex, education, income and region, with clustered standard errors at the individual level. Data are presented as estimated mean changes, with error bars indicating 95% confidence intervals. All statistical tests were two-sided.

## Treatment effects of misperception-correcting information on issue polarization

After estimating others' opinions about a given issue and being provided with the misperception-correcting information, participants were asked to re-report their own opinion about the target policy. This approach allows us to also examine whether providing misperception-correcting information reduces issue polarization—i.e., the difference in support for each policy between groups. Treatment effects on issue polarization (Fig. 3 and Supplementary Table 8) were of similar standardized magnitude to those estimated for affective polarization, though were less consistent. As hypothesized in preregistration, considering the bolsonaristas and lulistas group definition, we observed significant reductions in issue polarization relating to the abortion ($b = -0.24$, $t(1,798) = -3.83$, $p < 0.001$, 95% CI [$-0.36$, $-0.12$], standardized coefficient $= -0.04$) and Amazon conditions ($b = -0.22$, $t(1,798) = -2.60$, $p = 0.009$, 95% CI [$-0.39$, $-0.05$], standardized coefficient $= -0.05$). These effects were driven by significant reductions in lulistas' support for these policies (abortion: $b = -0.21$, $t(1,798) = -5.04$, $p < 0.001$, 95% CI [$-0.29$, $-0.13$], standardized coefficient $= -0.04$; Amazon: $b = -0.16$, $t(1,798) = -2.94$, $p = 0.003$, 95% CI [$-0.28$, $-0.05$], standardized coefficient $= -0.04$). Contrary to expectations, effects were not consistent for the racial and social quotas treatments. For the racial quotas treatment, increases in policy support emerged for both bolsonaristas ($b = 0.21$, $t(1,424) = 2.16$, $p = 0.031$, 95% CI [$0.02$, $0.39$], standardized coefficient $= 0.05$) and lulistas, though the latter was unsupported by Bayes factors ($b = 0.15$, $t(1,424) = 2.09$, $p = 0.037$, 95% CI [$0.01$, $0.30$], standardized coefficient $= 0.04$, $BF_{10} = 0.92$), leading to a non-significant effects on policy polarization ($b = -0.05$, $t(1,424) = -0.44$, $p = 0.662$, 95% CI [$-0.29$, $0.18$], standardized coefficient $= -0.01$), for which the Bayes factor corroborates this null result ($BF_{10} = 0.03$). The social quotas condition also had no significant impact on policy support (bolsonaristas: $b = 0.08$, $t(1,424) = 0.80$, $p = 0.425$, 95% CI [$-0.11$, $0.27$], standardized coefficient $= 0.02$, $BF_{10} = 0.07$; lulistas: $b = 0.07$, $t(1,424) = 0.98$, $p = 0.326$, 95% CI [$-0.07$, $0.21$], standardized coefficient $= 0.02$, $BF_{10} = 0.14$; issue polarization: $b = -0.01$, $t(1,424) = -0.08$, $p = 0.936$, 95% CI [$-0.25$, $0.23$], standardized coefficient $= -0.001$, $BF_{10} = 0.04$). Results are similar considering left-right ideological self-placement group definition and for people who hold anti-lulista and anti-bolsonarista sentiment (see Supplementary Figs. 22–23, respectively). Exploratory analyses found no evidence of heterogeneous treatment effects across waves 2 and 3 (abortion: $BF_{10} = 0.04$; Amazon: $BF_{10} = 0.06$). We also did not find significant heterogeneous treatment effects between waves 4 and 5 for racial quotas ($BF_{10} = 0.04$). Treatment effects were significantly different between waves 4 and 5 for social quotas ($b = 0.61$, $t(1,424) = 2.32$, $p = 0.021$, 95% CI [$0.09$, $1.13$] standardized coefficient $= 0.08$), but Bayes factors showed some evidence of a null effect ($BF_{10} = 0.75$; see Supplementary Fig. 24). Results remain largely unchanged as per a Benjamini-Hochberg correction for multiple hypotheses testing across alternative political groupings (Supplementary Table 9).

## Discussion

This study makes two main contributions. First, drawing on multiple definitions of political ingroups and outgroups, it longitudinally tracks affective polarization in a highly fractionalized multiparty system with low party identification, rampant personalism and anti-partisanship. Over five waves, using several ingroup-outgroup definitions, feeling thermometer evaluations were used to assess fluctuations before, between and after the October 2022 Brazilian elections, a period that had been widely anticipated to have a divisive influence on the country[42]. Our second contribution is the testing of misperception-correcting information in this context, where, for the first time in Brazil's current democratic period, an incumbent president competed against a former president, and both main candidates were considered highly controversial figures by non-supporters. Using the lulista-bolsonarista intention-to-vote group definition, the effect sizes of our four, alternative treatment conditions were, broadly speaking, similar in extent to the post-election softening of affective polarization recorded during the period of analysis (see Supplementary Fig. 25), with arguably the most morally-charged and salient of our controversial issues—abortion—associated with the largest treatment effects. This is noteworthy in relation to misperception-correction survey experiments fielded on various topics in the United States[22,23], and in light of a recent empirical assessment of dehumanization among political groups in Brazil, which concluded, "those who hope to reduce polarization in Brazil will have their work cut out"[24].

Brazil presents challenges to the measurement of affective polarization because of the multi-faceted varieties of political groups

of different definitional types that potentially warrant assessment. Existing measures in the literature that contend with the complexities of multiparty systems weight thermometer scores (the workhorse metric in the field[1]) by party vote share, and so cannot be straight-forwardly applied to this coalitional presidentialism system with frequent party switching, in which party identification is low and negative partisanship is common. Thus, published multi-country studies have so far either consciously omitted Brazil from international comparison[35] or applied party vote-share weighting schemas and assessed Brazil to have fairly low levels of affective polarization (lower, for example, than Canada and Portugal[16]), which fits poorly with Brazilianists' commentary[42]. The few single-country Brazil studies available use disparate assessments of affective polarization and sample the population differently, making comparison across political-group types difficult. Individually, they also take a narrower perspective of the potentially relevant political groups than that presented here, focusing either on citizens' affective evaluations of (non-)supporters of the Workers' Party[43], or on evaluations of supporters of a small number political parties or of the parties themselves[44,45], or on evaluations of left- and right-wing citizens' like and dislike scores towards communists and those who defend the military regime[46]. The aforementioned recent study of dehumanization reported larger gaps in gradings of how human (vs. how similar to lower animals) petistas vs. anti-petistas see each other, compared to lulistas vs. bolsonaristas[24]. This is in contrast with our feeling thermometer results, and may be explained by the other study excluding from pro-politician classifications respondents with overlapping − and thus perhaps more intensely experienced−identities (i.e., a lulista was defined a supporter of Lula who did not also support the Workers' Party, while a petista may or may not have supported Lula).

From the outset, we found that the lulista-bolsonarista grouping consistently gave the largest feeling thermometer interval, which remained fairly steady through the elections. The levels are similar to or somewhat higher than feeling thermometer intervals among partisans in the United States[22,23,47] and towards leaders and parties recorded in some other multiparty systems[35]. We also found a post-election reduction in lulista-bolsonarista affective polarization as per Reiljan and Ryan[48], and Hernández et al.'s multi-country study of post-election dynamics[49]. In response to the latter's call for future research, we report that the losing group (bolsonaristas, some of whom set up roadblocks and camps immediately after the results came out, and later stormed government buildings on 8 January 2023) had, on average, a steeper post-election reduction in ingroup evaluations relative to the winning group (lulistas), and a similar, rising gradient in outgroup evaluations (see Supplementary Fig. 7 for chart and statistics).

An advantage of being able to compare thermometer scores dynamically across different political group definitions is, in our case, that it may provide insight into the impact of negative campaigning, shown to be an important driver of affective polarization in the United States[47]. While analysis of the drivers of changing sentiments over time is beyond the scope of this paper, observations of aggregate dynamics shed light on the plausibility of possible causes of the post-election reduction for lulista-bolsonarista affective polarization. Changes in identity salience may be one of them[50,51]. For instance, the pro-politician grouping was the most salient political identity in wave 3 (between first-round and run-off elections, Supplementary Fig. 26), and was the only political group definition to show a strong reduction in salience after the elections (Supplementary Fig. 27). This pattern is in notable contrast to the anti-politician definition, which showed approximately steady salience through waves 2 to 5. Hence, while speculatory, an easing off of lulista and bolsonarista identities−be it due to the frequency of election news, political rallies, to social media activity, or some other cause[18,45]−appears more consistent with the patterns seen

than negative campaigning. Furthermore, had ideological polarization acted as a mediator[52] of the dynamics of lulista-bolsonarista affective polarization, a post-election easing in the strength of opinion that respondents registered towards a battery of policy-opinion questions would have been expected[13,14], but did not occur (Supplementary Figs. 12 and 28).

The literature has only a few dynamic assessments that inspect ingroup-outgroup evaluations over the course of an electoral period. Our survey timing allows for a particularly clean temporal analysis, with the first wave finishing before election campaigning was permitted, the third beginning the day after first-round elections and ending the day before run-off contests (full election results are announced a few hours after voting stations close in Brazil), and the last round fielded after the victor took office. Aside from this study, multi-wave surveys to track affective polarization sub-annually, both before and after voters go to the polls, have been conducted during patches of the recent political history of Israel (10 waves from 2019 to 2021, covering four national elections[53]) and Spain (four waves from late 2018 into 2019, with three of these waves scheduled alongside a combination of regional, national and European elections[54]). There is also a six-wave study of the brief period up to the Danish parliamentary elections of 2011, which were called (endogenously) on the prerogative of the prime minister and held less than three weeks later[55]. Often due to data limitations, assessments of affective polarization's dynamics around elections sometimes rely on variation in the proximity between survey-response dates and election dates[56], using responses within a single survey wave collected close to an election. This approach can reveal substantial changes−for example, Lior Sheffer reports that ingroup bias dropped by a third just two days after the 2015 Canadian federal elections[52]−but inevitably constricts the period of analysis within that which may be empirically relevant. Overall, existing electoral-period studies generally make the case that dynamic fluctuations in affective polarization are larger around elections than between them[14].

Therefore, if dynamic measures of affective polarization and associated phenomena place correction effect sizes in context−since temporal fluctuations potentially overwhelm experimental corrections within a month or two during calmer times than elections (as Dias et al. recently argued in a meta-analysis of misperception-correction treatments tested in the United States[23])−then the similarity between treatment effect sizes and election-period change in this study may offer a little hope. To date, many types of treatments to reduce affective polarization have been tested in the U.S. context, from training followed by door-to-door canvassing in out-partisan areas[57], to cross-partisan video calls, to variations on brief, simple messages that are low-cost as well as scalable, among which, misperception corrections are considered to have minor and variable effects, if any[22]. For example, among interventions evaluated in Voelkel et al.'s megastudy, a "correcting policy misperceptions chatbot" yielded a Cohen's d of 0.16 ($p < 0.001$), and a treatment about "party overlap on policies", a Cohen's d of 0.17 ($p < 0.001$)[22]. The average partisan animosity reduction in Dias et al.'s misperception-correction meta-analysis was 2.6 percentage points, and the authors' point out that more effective misperception corrections are often those that deliver sequential interventions[23,58]. Our most effective treatment condition was about abortion, yielding a 6.66 percentage-point reduction in liking (Cohen's d = 0.26) and a 7.18 percentage-point reduction in perceived intelligence (Cohen's d = 0.28), and, as per all treatment conditions tested here, it was a brief, single, item.

How might our modestly larger effect sizes than found for many other policy-stance misperception corrections be explained? We deliberately considered controversial policy issues, in light of indications in the literature that they may be particularly useful for assuaging outgroup animosity. For example, Ryan[59] draws on the psychology literature, to argue that moral attitudes, especially those for which

views are deeply-held and opposing, engage "a distinctive mode of processing" because they "arouse certain negative emotions". Similarly, moral reframing is known to be highly effective at shifting support for and against political candidates[60]. Taking the argument further, most definitions of the word "controversy" entail not only opposing views that tap into moralized issues, but also an element of commonplace interest or widespread discussion. If a topic is particularly prominent, then some increased accessibility is implied, suggesting enhanced cognitive processing of messages relating to these issues[32,33]. Our most effective treatment condition, policy stance on abortion rights in the first three months of pregnancy, like the other conditions explored here, appeals to moral foundations additional to that of loyalty/betrayal[61–63]. Indeed, abortion is often treated as an archetypal moral issue in the literature; in the United States, it is frequently used to study the antecedents[64–66] and consequences[67] of attitude moralization (the extent to which people perceive attitudes to be imbued with moral convictions). In Brazil, moral values have been demonstrated to underpin physicians' willingness to work in the area of abortion, even in circumstances where it is straightforwardly legal[68]. When we then turn to consider commonplace interest or widespread discussion, an easily available indicator is the relative frequency of Google searches. Upon restricting search data to Brazil, we find that abortion has also been searched for consistently more often than the other policy issues used in our treatment conditions (see Supplementary Fig. 29). Our second-most effective treatment, Amazon deforestation, is the second-most commonly searched for topic.

Changing people's views towards sociopolitical groups or towards policies using misperception correction involves providing an external signal strong enough to overcome the internal psychological costs of change. On the one hand, controversial issues are associated with entrenchment of views, as established in the persuasion knowledge literature[69], implying high costs of changing one's attitudes. However, on the other hand, that same literature argues that group stances on controversial issues are also likely to be strongly diagnostic, and thus highly informative[31,70]. Thus, the patterns of effect sizes that we find for our (i) affective polarization and (ii) issue polarization outcomes may reflect the interplay between the variable signal strengths of the different misperception-correction conditions, and respondents' individual psychological costs of (i) adjusting their degree of like (or dislike) and perceptions of intelligence towards sociopolitical groups, and (ii) of revising their stances on policies. Respondents may have especially entrenched views on the issue of abortion, for example, but learning that they wildly mischaracterize outgroup opinion on this issue may especially strongly prompt re-assessment of outgroup likability. This suggestion is an area for future research, which we encourage for policy misperception corrections. Although these may not generally be as effective at reducing partisan animosity as some other approaches, they have a normative advantage beyond the practical considerations of cost and scalability: for those who wish to encourage programmatic politics, they shift voters' attention to substantive policy issues, as opposed to groups' demographics, for example[43]. We also encourage other researchers to contribute to knowledge about the effectiveness of experimental treatments to reduce affective polarization outside of the United States context.

This study has a number of limitations, among them the possibility of social desirability bias affecting survey responses[71], though this concern is more pertinent for our issue polarization treatment results than our affective polarization findings, because only in the former case are our treatments similar in content to downstream attitude assessment[23]. While we aimed at a representative sampling of the Brazilian population in terms of sex, age and region, income and educational attainment, as per the Brazil's census, our final sample under-represents individuals with lower levels of schooling (as our surveys required reading skills). The practical considerations of maintaining a representative, four-wave panel places constraints on

sample size, and hence limitations on statistical power that did not permit subgroup analysis of treatment effects. Our design was also not intended to assess long-term effects or the examination of order effects across all conditions. Finally, although we ask people to evaluate groups of citizens rather than leaders or parties, and the semantics of Portuguese terms like "lulista" and "anti-lulista" help with this[72], we cannot be certain that some respondents do not call to mind political elites (i.e., Lula rather than his supporters) when asked about horizontal sentiment.

Noting these limitations, the current work demonstrates both different levels and dynamic variation of affective polarization across political group definitions in Brazil, and that information intended to correct misperceptions of group positions on controversial policies effectively mitigates outgroup animosity during a tense electoral period. We propose that exploring the controversial aspect of some policy issues may provide a source of optimism for those seeking light-touch, short, simple, policy-focused interventions. Our broader hope is that this study and the dataset supporting it contribute to the field's evolving understanding of how to assess affective polarization in societies where political affiliation is multifaceted, and of how to sustain normatively positive political distinctions alongside empowered citizen choice[73], whilst keeping in check less desirable aspects of elections as central institutions in democratic polities.

## Methods

### Multi-wave surveys

We collected data over five waves starting at (1) April 2022 (2) early July 2022; (3) October 2022; (4) late-November 2022; (5) late-January 2023. Waves 1 and 2 occurred before the elections (April and July 2022, respectively), wave 3 between the first round of the elections and run-offs (October 2022) and waves 4 and 5 after the elections (wave 4, during the World Cup, and wave 5 shortly after Lula took office, and subsequent rioting in the capital). The data collection for an additional wave, pre-registered for mid-August 2022, did not take place due to operational challenges. Even though we pre-registered assessment of fluctuations in affective polarization as the elections approached, we were also interested in how it behaves after the elections. Participants of all waves were members of the Netquest Brazil panel, which includes approximately 140,000 active members.

We established quotas for subnational regions (Northeast, Southeast, South and North/Center-west), age, sex and family income, with sample sizes per quota proportional to the participation of each stratum in the Brazilian population. While we tried to keep the sample representative of the Brazilian population in terms of educational attainment, there was some under-representation of low educational levels, which is to be expected in an online panel that requires the reading of questions on electronic devices (see details of sample composition in Supplementary Table 2). The first wave included 2,030 respondents. The second wave included 2,931 participants (with some overlap with the first wave for strata with smaller numbers of participants in the Netquest panel). This wave deliberately exceeded the sample size required to achieve 80% power to detect small effect sizes across our primary hypotheses, in order to accommodate anticipated attrition across subsequent waves. Waves 2, 3 (2,314 participants), 4 (1,938 participants) and 5 (2,104 participants) composed a longitudinal panel including repeated measures for respondents of wave 2. All waves included 3 attention checks and respondents that failed all were excluded from analysis (0.88% in wave 1, 0.00% in wave 2, 0.00% in wave 3, 0.64% in wave 4 and 1.17% in wave 5). We also excluded incomplete responses.

### Assessing polarization

**Political identification definition.** We assessed affective polarization as the difference in the evaluations of ingroup and outgroup targets (as

defined by political self-identification) by respondents, in terms of liking and perceived intelligence. We measured issue polarization as the difference in policy opinions between different political groups. The survey included vertical as well as horizontal affective polarization questions, though only the latter were used in this study.

We measured political self-identification in three ways for statistical assessment throughout the study:

POL1 - Left-right continuum on a 5-point scale (1-clearly left-wing, 2- left-wing, 3-center, 4-right-wing and 5-clearly right-wing; 90-"not identified" and 99-"don't know".)

POL2 - Identification with candidates' political groups. 1- lulista (intentions to vote for Lula in first round of presidential election), 2- bolsonarista (intentions to vote for Bolsonaro in first round of presidential election), 3- neither.

POL3 - Aversion to politicians' political groups: 1- anti-lulista (would never vote for Lula), 2 - anti-bolsonarista (would never vote for Bolsonaro), 3- anti-lulista and anti-bolsonarista (would never vote for either) and 4- neither.

In Wave 1, we also recorded ratings of supporters of the Workers' Party ("petistas") and those who say they would never vote for that party ("anti-petistas"). However, we dropped this assessment after first-wave data showed that these political group definitions were less common as well as less salient political identities than others. Based on our three main self-identification measures, response categories were grouped to describe ingroups and outgroups as follows: POL1: 1,2 (clearly left-wing and left-wing) vs 4,5 (right-wing and clearly right-wing); POL2: 1 (lulista) vs 2 (bolsonarista); POL3: 1 (anti-lulista) vs 2 (anti-bolsonarista). We subsequently created ingroup and outgroup correspondences as per those categories, and calculated affective sentiment expressed towards targets (e.g., lulista respondents with evaluation of lulistas and bolsonaristas targets). For the ideological left-right identification grouping, participants evaluated left- and right-wing individuals who talk a lot about politics and talk little about politics. We averaged these characterizations to have a single ingroup and a single outgroup evaluation.

To measure salience, we asked respondents to rank the three most important items for their own sense of identity, from among a list that they had just answered self-categorization questions about (age, sex, educational attainment, income, region of residence and political affiliation). Based on responses to this question, we created a dummy variable indicating whether or not the participant included political affiliation. We randomized between-subjects which measure of political self-categorization was shown before the identity-salience question, with the others presented afterwards (lulista/bolsonarista, anti-lulista/anti-bolsonarista, left/right wing, petista vs. anti-petista).

**Ingroup and outgroup evaluations.** Evaluations of ingroups and outgroups were collected for two different variables: liking and intelligence. Participants evaluated targets from categories of different self-identification political groupings described above, in each case referring to groups of citizens of different political persuasions, rather than to elites. Depending on the respondent's political self-identification, responses are coded as ingroup (same label of self-identification and evaluated target identification) or outgroup evaluation (opposed self-identification and target identification). For example, for a left-wing respondent, the evaluation of left-wing people (target identification) is considered as ingroup evaluation, while the evaluation of right-wing people is considered outgroup evaluation. Thus, four different variables were created, with ingroup and outgroup evaluations being repeated measures for each subject.

OE1i: Liking of ingroup (0 – 100 continuous scale) OE1o: Liking of outgroup (0 – 100 continuous scale) OE2i: Intelligence of ingroup (0 – 100 continuous scale) OE2o: Intelligence of outgroup (0 – 100 continuous scale)

We also collected meta-evaluations for all combinations of ingroups and outgroups (e.g., how do you think that left-wing people feel about right-wing people, as well as the other 3 combinations: right-wing about left-wing, left-wing about left-wing, and right-wing about right-wing). Responses of ingroup and outgroup meta-evaluations are averaged (e.g., an ingroup meta-evaluation was the average of (i) evaluation of left-wing individuals by left-wing individuals, and (ii) right-wing individuals by right wing-individuals), so as to have only two measures: perception of meta-evaluations of ingroups and outgroups. Thus, four different variables are used, two for intelligence and two for liking, each with ingroup and outgroup evaluations being repeated measures for each subject.

OM1i: Perception of levels of liking between in-groups (0 – 100 continuous scale) OM1o: Perception of levels of liking between in-groups and out-groups (0 – 100 continuous scale) OM2i: Perception of levels of intelligence between in-groups (0 – 100 continuous scale) OM2o: Perception of levels of intelligence between in-groups and out-groups (0 – 100 continuous scale)

We collected evaluations of policy support for many different policies across all survey waves, including:

OP1b: Support for abortion before 3rd month of pregnancy (7-point scale, before treatment)

OP2b: Support for zero deforestation of the Amazon (7-point scale, before treatment)

OP3b: Support for racial quotas (7-point scale, before treatment)

OP4b: Support for social quotas (7-point scale, before treatment)

And we collected a second measure of the same evaluations - after treatment - for policies considered in the misperception-correction treatments:

OP1a: Support for abortion before 3rd month of pregnancy (7-point scale, after treatment in waves 2 and 3)

OP2a: Support for zero deforestation of the Amazon (7-point scale, after treatment in waves 2 and 3)

OP3a: Support for racial quotas (7-point scale, after treatment in waves 4 and 5)

OP4a: Support for social quotas (7-point scale, after treatment in waves 4 and 5)

## Demographic and other socio-political variables

Following the approach of the Brazil's census, we collect demographic information (age, sex, educational attainment, income, region of residence), as well as self-categorization for various political groups, as described above. We also collected data on views of nationalism and perceptions about the political meaning of the national football team shirt. We ask respondents to self-define their race using the same terminology and the same five options available in the Brazilian national census of 2022 (the most recent), with the additional option of "prefer not to say".

## Misperception-correcting information treatments

In wave 1, we collected baseline data on policy support that was used in subsequent waves as part of the misperception-correcting information treatments. From wave 2 onwards, randomly assigned treatments were included in the survey. As described in Results, in wave 2, respondents were randomly assigned to a control condition, or to a misperception-correcting information treatment about an abortion policy, or to a treatment about a policy of zero deforestation in the Amazon. In wave 3, participants who had been in the control condition in wave 2 were again assigned to the control condition in wave 3. Those who had been in the abortion and Amazon treatment conditions in wave 2 were counterbalanced (i.e., respondents who had been assigned to the abortion condition in wave 2 were then assigned to the Amazon condition in wave 3, and vice-versa). In wave 4, participants were randomly assigned (independently of their assignment in waves 2 and 3) to either the control condition, or to the social quotas or racial quotas

conditions. Analogous to the transition from wave 2 to wave 3, in wave 5, those who had been assigned to control in wave 4 were again assigned to control, and those who had been assigned to the social quotas and racial quotas conditions, respectively, in wave 4, were counterbalanced (see Supplementary Table 3).

For each treatment, participants were first introduced with a brief and general text about the current legislation on the matter and were told that people had different opinions (see Supplementary Table 1). They were then asked to estimate, in random order, how many bolsonaristas and lulistas out of 10 supported the policy in question. They were then presented with the actual policy support data for both groups from wave 1 (see Supplementary Table 1).

To test hypotheses about misperception-correcting treatments effects on affective polarization, a 2 (evaluated group: ingroup vs outgroup) x 3 (content: control vs treatment 1 vs treatment 2) design was used. To test hypotheses about effects on issue polarization, a 2 (order: before vs after treatment, within subjects) x 2 (political self-identification: lulista vs bolsonarista, between subjects) x 2 (policy: treatment 1 vs treatment 2, between subjects) mixed design was used, with replicates for other political identification groupings (left vs right, and anti-lulista vs anti-bolsonarista). To assess our main treatment effects, we combine data from waves with identical treatment conditions to estimate the average effects of each condition (rather than performing simple comparisons of means over separate waves). For example, the assessment of the Amazon condition combines data from waves 2 and 3.

The study was pre-registered in AsPredicted as a series of related projects, including: a) investigating changes in polarization during the election and treatment effects relating to abortion and Amazon conditions in wave 2 (AsPredicted #102137, 9 July 2022, https://aspredicted.org/RSC_3NW) and wave 3 (AsPredicted #108708, 5 October 2022, https://aspredicted.org/ag5nx.pdf), b); assessing treatments associated with affirmative action policy in waves 4 and 5 (AsPredicted #113790, 21 November, https://aspredicted.org/6LH_XWW), and c) investigating changes in polarization during the World Cup in wave 4 (AsPredicted #114066, 5 November 2022, https://aspredicted.org/c8xm2.pdf).

Hypotheses of pre-registration #102137 and # 108708 (dynamics of polarization over time and Amazon/abortion treatments):

Hypotheses H1a: Over time, as elections approach, the difference between ingroup and outgroup evaluation (liking and intelligence) will increase.

H1b: Over time, as elections approach, the difference between ingroup and outgroup meta-evaluation (liking and intelligence) will increase.

Hypotheses H2a: Presenting information about counter-stereotypical support for expansion of abortion rights policy will reduce the difference between ingroup and outgroup (as defined by political self-identification) evaluations (liking and intelligence).

H2b: Presenting information about counter-stereotypical support for zero deforestation of the Amazon policy will reduce the difference between ingroup and outgroup (as defined by political self-identification) evaluations (liking and intelligence).

H3a: Presenting information about counter-stereotypical support for expansion of abortion rights policy will reduce the difference between ingroup and outgroup (as defined by political self-identification) meta-evaluations (liking and intelligence).

H3b: Presenting information about counter-stereotypical support for zero deforestation of the Amazon policy will reduce the difference between ingroup and outgroup (as defined by political self-identification) meta-evaluations (liking and intelligence).

H4a: Presenting information about a group's counter-stereotypical support for expansion of abortion rights policy will reduce the difference in support for this policy between opposing groups (as defined by political self-identification).

H4b: Presenting information about a group's counter-stereotypical support for zero deforestation of the Amazon policy will reduce the difference in support for this policy between opposing groups (as defined by political self-identification).

Hypotheses of pre-registration # 113790 (affirmative action treatments)

H1a: Presenting counter-stereotypical information about support for a policy that provides access of low-income families to higher education by means of quotas (i.e., information that most right-wing people do not oppose these quotas) will reduce the difference between ingroup and outgroup (as defined by political self-identification) evaluations (liking and intelligence).

H1b: Presenting counter-stereotypical information about support for a policy that provides access of Afrobrazilian and indigenous students to higher education by means of quotas (i.e., information that almost half of right-wing people do not oppose these quotas) will reduce the difference between ingroup and outgroup (as defined by political self-identification) evaluations (liking and intelligence).

H2a: Presenting counter-stereotypical information about support for a policy that provides access of low-income families to higher education by means of quotas will reduce the difference in support for this policy between opposing groups (as defined by political self-identification).

H2b: Presenting counter-stereotypical information about support for a policy that provides access of Afrobrazilian and indigenous students to higher education by means of quotas will reduce the difference in support for this policy between opposing groups (as defined by political self-identification).

To test hypotheses related to changes in polarization over time, we use linear regression models with dummy variables representing waves (or aggregated in pre, mid- and post-election periods) controlling for demographics and treatment conditions. We perform robustness checks considering only participants in the control condition. To test randomized treatment effects, we used data from waves 2–5, employing respondent fixed-effects rather than demographic controls to further reduce error variance. While we acknowledge the limitations of fixed-effects models in disentangling between and within-subject variation[41,74], these concerns are not critical in our case as our predictors of interest are exogenous (temporal proximity to the election and randomly assigned treatment conditions), ruling out potential omitted variable biases by design. Nonetheless, we perform robustness checks with mixed-effects regression.

Concerning pre-registration #102137, to test H1a, two linear regression models were estimated. The dependent variables of each model were OE1, and OE2 and the predictors, dummy variables representing waves 2, 3 and 4 (and for parsimony an alternative model aggregating waves into pre-election (waves 1 and 2), election (wave 3), and post-election (waves 4 and 5), a dummy variable indicating if the evaluation refers to an ingroup (1) or outgroup (0) and their interaction. To test H1b, a similar model had OM1 and OM2 as alternative dependent variables. Treatment conditions are orthogonal to time and are used as covariates to increase efficiency of the estimators. All models will include as controls subnational region, age, sex, education and income (which will also be used to generate quotas representing Brazilian population for data collection) and clustered standard errors at the individual level. Each model was estimated for three different self-identification groupings (POL1, POL2, and POL3). H2a, H2b, H3a and H3b were tested using a regression model including as predictors a dummy variable indicating if the evaluation refers to an ingroup (vs. outgroup) and its interactions with dummy variables representing abortion and deforestation treatment conditions, as well as individual fixed-effects (which capture all variability between subjects, hence only interactions with treatment effects will be included, but not their main effects).

For robustness, additional models including as covariates the interaction of outgroup with age, sex, interest in politics, religiosity and educational level were estimated. Standard errors were clustered at the individual level to account for dependency across repeated measures. Each model was estimated for 3 different self-identification groupings (POL1, POL2 and POL3). Data from waves 2 and 3 (that had the abortion/Amazon treatments), as well as data from waves 4 and 5 (that had the social and racial quotas treatments) were analyzed jointly alongside a model with interactions of treatment and wave. A regression model similar to the one used to test treatment effects, with additional interaction terms with a dummy variable representing the wave, was used to evaluate differences in treatment effects across waves (i.e., the difference between wave 1 and 2 for abortion/Amazon treatments, and the difference between wave 3 and 4 for social/racial quotas treatments).

To test H4a and H4b, two linear regression models were used, with policy support as a dependent variable (for OP1 and OP2) and, as predictors, a dummy variable indicating post-treatment response, a dummy variable indicating the respondent's self-identified political group (for POL1, 1= left-wing, for POL2, 1=lulista, for POL3, 1=anti-lulista), and their interaction. Each model was estimated for POL1, POL2 and POL3.

We estimated treatment effects using linear models with cluster-robust standard errors (clustered at the respondent level). Cluster-robust variance estimators remain consistent under heteroskedasticity and within-cluster correlation, and do not require normally distributed errors for valid large-sample inference[75–78]. With a large number of clusters (as in our case, with over 1000 respondents), the asymptotic approximations underlying cluster-robust inference are well supported[78]. Relevant model assumptions include exogeneity of predictors and independence between clusters. Exogeneity is ensured through two design features: (1) treatment conditions were randomized, guaranteeing independence of treatment assignment from potential outcomes and covariates in expectation, and (2) the timing of electoral periods (pre-, during-, and post-election) represents pre-determined temporal variation that is independent of individual-level potential outcomes[79,80]. We assume independence across clusters, which is satisfied because respondents were independently recruited online with no systematic interaction between clusters. Any remaining within-cluster correlation is accommodated by the cluster-robust variance estimator[76,78].

A Benjamini-Hochberg correction of p-values testing was used to correct for multiple hypotheses testing across different self-identification groupings. Analyses were performed using Stata version 18.0.

### Procedures for calculating Bayes factors in the direction of the alternative hypothesis (BF$_{10}$) and equivalence tests

We employed both classical and Bayesian methods to analyze our data. While classical analyses offer widely recognized metrics such as p-values and confidence intervals, these frequentist statistics do not allow for quantifying evidence in favor of the null hypothesis. Bayesian methods, however—particularly through the use of Bayes factors—do allow us to evaluate support for both null and alternative hypotheses[81]. Yet because these two approaches rely on distinct statistical assumptions (for example, the Bayesian framework incorporates prior distributions, while classical inference relies solely on sampling distributions), the results obtained are not directly comparable. We consider the classical findings as the basis for our conclusions, and any convergence or divergence between them should be interpreted in light of their fundamental differences. Nonetheless, this dual approach enables us to validate our findings across methodological frameworks, ensure robustness, and address the interpretive preferences of diverse audiences.

For the Bayesian analyses, we followed the same approach as in Johnston and Madson[82] by fitting a Bayesian generalized linear model and calculating the model parameters of interest. For each coefficient, we used the Savage–Dickey density ratio to determine the change in the probability of the null hypothesis from the prior to the posterior distribution by comparing their densities at the null value. Both independent and dependent variables were standardized, and we started with a normal prior distribution for all coefficients with a mean of zero and a standard deviation of 0.50 as considered in previous studies about political ideology[82]. For all coefficients of interest, we then computed Bayes factors for non-directional hypotheses (two-tailed) using the R package bayestestR[83]. We also report qualitative interpretations of the estimated Bayes factors based on Andraszewicz et al.[84]. For equivalence tests, we conducted one-sided tests (TOST)[85] with the alternative hypothesis that the absolute value of the effect in the population is smaller than the smallest effect size of interest (SESOI). Following pre-registered effect sizes of previous work[86] that similarly examined reductions in affective polarization, we set the SESOI for all equivalence tests at d = 0.16.

### Ethics and inclusion

The study was approved by the Oxford University Central Research Ethics Committee (SSH/BSG_C1A-22-11; SSD/CUREC1A/BSG_C1A-22-11/ amendment 01). All study participants provided informed consent, and no minors were involved. All authors bar one (AP) are Brazilian. The research process was conducted so as to support the career development of junior team members, and to take into account locally produced, relevant prior research.

### Reporting summary

Further information on research design is available in the Nature Portfolio Reporting Summary linked to this article.

## Data availability

Anonymized data used in this study is available at https://doi.org/10. 17605/OSF.IO/Y8XKM.

## Code availability

Anonymized code for replicating the findings of this study is available at https://doi.org/10.17605/OSF.IO/Y8XKM.

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

## Acknowledgements

The surveys were funded through the Lemann Foundation Program at the Blavatnik School. The authors thank A. Reiljan for feedback on an early draft.

## Author contributions

A.P., G.R., R.G. and E.A. conceptualized the study and developed the survey. R.G., G.R. and R.F. conducted statistical analysis. A.P., E.A. and R.G. wrote most of the main manuscript, with important ideas, written contributions and edits from G.R. and R.F. All authors approved the final version of the manuscript for submission.

## Competing interests

The authors declare no competing interests.
