## [Transparent Peer Review file · Nature Communications]

Electoral dynamics of sociopolitical polarization in Brazil across multiple political-group definitions with misperception-correcting treatment effects

Corresponding Author: Dr Anna Petherick

Version 1:

Reviewer comments:

Reviewer #1

(Remarks to the Author)

I enjoyed reading this timely research examining longitudinal data on polarization, meta-perceptions, and corrective interventions in the context of the 2022 Brazilian election. This research represents a substantive methodological and theoretical contribution to our knowledge of (meta)polarization. I applaud the use of preregistered panel data, equivalence testing, and examinations of a theoretically distinct political context. Below I lay out several comments in the hope that they will be useful to the authors as they revise their manuscript.

Perhaps my largest suggestion relates to clarity throughout the manuscript, not only in technical details (methods, analyses), but in also clarifying the contributions. I'll preface by noting that I'm familiar with the difficulty of how to include certain information in the Nature Journal format, where methods come at the end. So some of what I'm looking for is merely a matter of mentioning crucial information earlier, when possible, but some of the information I'm requesting isn't detailed enough anywhere. First, I'd strongly suggest more information about the analyses and modeling choices. There are myriad ways to analyze panel data, ways that are often at odds in terms of what they're trying to isolate, and ways that have long been standard but have now come under strong criticism (e.g., Bell & Jones, 2015; Lucas, 2023). There is a dearth of information in the manuscript about how the analyses are conducted. From what I can glean from the regression tables the authors are using simple Fixed-effects models, but as Bell & Jones (2015) note, these models do not appropriately isolate within-person longitudinal effects. Put another way, I would seem the presented analyses are looking primarily at mean differences between waves or between-subjects differences collapsed across waves. Both of those choices collapse across between-person differences and within-person changes over time (see Curran & Bauer, 2011). As such, I would suggest the author consider alternative modeling strategies that disaggregate within- and between-person effects. Doing so exploits the full value of having longitudinal data, and I think the paper is currently leaving a lot of useful analyses unexamined. Equally important, I would ask that the manuscript provide clear information about the panel-modeling strategies being employed, and how those choices appropriately match the stated hypotheses.

There are also several details about the interventions themselves that are unclear in the main text, or not discussed at all. One thing that is unclear in the main text is how the intervention is deployed/randomized, for example are people randomized to treatment vs. control at each Wave, or only at Wave 2, from which point that assignment is static? At times it sounds like the effects of the intervention are measured across waves (e.g., intervention at W3, measure at W4), other times it seems like they're at the same wave. It's also unclear whom participants are getting corrective information about, at points it sounded like participants are getting information about ingroup attitudes, outgroup attitudes, and/or general population attitudes. This all needs to be clearer in the main text of the manuscript.

I would also suggest the paper do more to spell out the theoretical and methodological contributions of examining polarizations and meta-perceptions in such an atypical political system/culture. What are the implications for intergroup theory precisely (e.g., how do the findings challenge past theorizing that is built on data from only US/European political contexts), and what are the methodological implications? On this latter point I think the manuscript as written undersells itself. I found their measurement and operationalization(s) of identity/political boundaries quite novel, but one only realizes this after reading the methods section. While reading the main text I had a reaction of "well you keep telling me this is a political context that is theoretically distinct from places like the US, but you're still using a binary in/out-group boundary

operationalization like researchers in the US do." It was only upon reading the methods that I truly appreciated the immense thought the authors put into how they operationalized these identities, and how what they did makes a significant theoretical and methodological contribution. I would suggest that authors better foreground these contributions in the manuscript! Personally, if I were someone considering how to study polarization in a political context unlike the US, I'd want to use the methods these authors used here!

Minor Comments:

- The preregistrations state discrete, numbered hypotheses. I would suggest referring to these numbered hypotheses in the manuscript.
- Is there a formal test of the (equivalency) claim that the interventions were similar in size to the post-election drop? This is something claimed in the title of the paper, but if there's a formal test of the equivalence in effect size, I missed it.

I hope the author(s) find my comments helpful as they revise their manuscript.

Cited:

Bell, A., & Jones, K. (2015). Explaining fixed effects: Random effects modeling of time-series cross-sectional and panel data. *Political Science Research and Methods*, 3(1), 133–153. <https://doi.org/10.1017/psrm.2014.7>

Curran, P. J., & Bauer, D. J. (2011). The disaggregation of within-person and between-person effects in longitudinal models of change. *Annual Review of Psychology*, 62(1), 583–619. <https://doi.org/10.1146/annurev.psych.093008.100356>

Lucas, R. E. (2023). Why the cross-lagged panel model is almost never the right choice. *Advances in Methods and Practices in Psychological Science*, 6(1), 25152459231158378. <https://doi.org/10.1177/25152459231158378>

(Remarks on code availability)

The code is Stata code, and I do not have Stata nor do I know how to use it. I use R exclusively.

Reviewer #2

(Remarks to the Author)

In the paper "Sociopolitical Polarization Reductions After Elections Are Similar in Extent to Correcting Misperceptions about Controversial Policies" the authors investigate the dynamics of affective polarization before and after the Brazilian election and whether providing information on the beliefs of the outgroup can reduce polarization.

While the paper provides some interesting results, I don't think that the results are sufficiently significant to warrant publication in a high-impact journal like *Nature Communications*. The main take-aways from the paper are that there is high affective polarization between Bolsonaro supporters and Lula supporters, that this polarization decreased somewhat after the election and after correcting misperceptions about the beliefs of the outgroup, and that the FIFA world cup had not meaningful effects on polarization. None of these effects are very surprising and although they provide an addition to the literature, they are not very novel. The most interesting effect in my opinion is that polarization decreases after providing information about the outgroups' beliefs about controversial issues. While certainly interesting, this effect is only an extension of earlier work (1), as the authors themselves note.

On the positive side, the study reported in the paper as well as the analyses seem very sound and are well-reported.

A smaller technical issue for me was the equivalence tests the authors used. I'd recommend using Bayes factors to test for absence of effects as these don't rely on relatively arbitrary decisions about the smallest effect size of interest.

Another recommendation would be to drop the analyses on the impact of the FIFA world cup on polarization. In my view, the rationale behind expecting effects of the event on polarization is rather weak and the results are also not very noteworthy. I think dropping this analysis would improve the readability and focus of the paper.

References

(1) Voelkel, J. G. et al. Interventions reducing affective polarization do not necessarily improve anti-democratic attitudes. *Nat Hum Behav* 7, 55–64 (2022).

(Remarks on code availability)

I could not access the code, because I needed permission.

Reviewer #3

(Remarks to the Author)

Review *Nature Comms*

This paper assesses changes in the levels of affective polarization in Brazil in the context of the last presidential election in this country. The papers use a four-panel wave conducted before, after, and during the election. This data is augmented with experimental instruments embedded in the surveys to measure the effects of correcting misperceptions of ingroup and outgroup options on affective polarization. The paper is well-written, well-motivated, and carefully developed. With that said, I have suggestions to offer in the structure of the paper and the methodological approach.

Framing

In the structure of the paper, in my view, the descriptive content about the stability of the levels of polarization, and the null effects of the world cup and the January 8th riots are the least exciting components of the paper. As a matter of fact, in my view, the authors spend too much time discussing these dynamics, when they merely serve the purpose of showing that polarization in Brazil is stable and hard to change. This is the main message of the paper's introductory section, which can be presented in a single section instead of in three results sections.

The decision to have the World Cup and the riots as separate sections of the paper distracts the reader from the most exciting results of the paper: the effects of correcting misperceptions on levels of polarization. Even though the authors bring novel data, the fact that Brazil is a polarized country has been documented already (as some of the literature cited by the authors show, Zucco and Samuels' work and Ortelado, and more recently work by Nunes). However, serious policy evaluations of interventions to reduce are still more scarce, and I think this is where the paper's main contribution resides. So I suggest the authors move these two sections to the end of the paper or incorporate them as smaller paragraphs in the first results section.

At the same time, with this reorganization, the paper will become more about measuring interventions to reduce polarization. So the authors will need to give higher centrality to this topic in the introduction of the paper. This is already there, but the authors must flash out a bit more.

Methods

Regarding the methodological aspects, I have some questions about the models used in the experimental section. The authors mention their treatment is an intervention to correct misperceptions. So on average, voters believe X people to support policy Y , but the actual number is that on X is smaller than what the average voter believes. While on average, the treatment is correcting misperceptions, the treatment is actually doing different things conditional on where the voter i is in the full distribution.

In the current form, the authors present the results by treating their intervention as an intention-to-treat effect. This is fine, and I think the authors should keep this available for readers as the primary analysis. However, the authors have much richer data, and I would like the author to provide extra analysis that could be incorporated into the paper using a few different options to measure the treatment effect.

First, I would like the authors to present these results separating by subgroup between those in which the treatment did exactly what the authors designed for theoretically (reduce misperceptions), and those who already had a view close to or smaller than the actual number.

This can be done by splitting the data, or the authors could estimate the Complier Average Causal Effect using a IV setup. Splitting the data seems actually a superior choice because this is not a matter of compliance, but more of the treatment potentially be pushing respondents in different directions.

Lastly, I would also be interested in some type of continuous measure of the correction. For example, I imagine the effects are much larger, given how wrong I was about an outgroup. This is important to be added to the paper.

Still in the methods section, these different analyses could also allow the authors to speak to the issue of social desirability bias. It is hard to disentangle if the experimental results come from the fact that respondents are adjusting their polarization levels because they hold more accurate views about the outgroup, or because the survey is just simply correcting them. Looking at the sensitivity of the results conditional on the gap and the subgroups could also allow the authors to speak to this issue.

I hope these comments help the researchers improve the paper.

(Remarks on code availability)

Version 2:

Reviewer comments:

Reviewer #1

(Remarks to the Author)

I enjoyed reading this revised manuscript examining polarization/misperception correction interventions in Brazil. I am very stratified with the authors' responses to my original comments and concerns. The inclusion of the information and clarifications that I asked for is good, and I find their defense of the use of fixed-effects panel modeling convincing. Moreover, I was happy to see the foregrounding of what I (still) believe is the primary contribution of this work: a gold standard model for conducting polarization and misperception reduction interventions outside of binary political-identity contexts like the United

States, which dominates research and theorizing in this literature. I don't use the term gold standard here lightly. This paper is not merely generalizing past findings or slightly modifying existing theory and method, it's presenting a carefully thought out and integrated theoretical and methodological framework for generalizing (and challenging) a literature that is, in my opinion, deeply hampered by its US-centrism.

I find the paper well written, the research exceptionally thorough and expertly conducted, and the potential for significant contributions high.

(Remarks on code availability)

Reviewer #2

(Remarks to the Author)

The authors addressed all my previous comments in a satisfactory manner. I'm still not convinced that the results of the manuscript are novel enough to warrant publication in Nature Communications but I leave it to the editor to decide on that matter.

(Remarks on code availability)

Reviewer #3

(Remarks to the Author)

I appreciate the work the authors put into revising the manuscript. All my recommendations were thoroughly responded, and I am satisfied with the authors' responses. I believe this paper meets the requirements for publication at Nature Communications

My only remaining suggestion is for the authors to incorporate the results using the alternative specifications for misinformation intervention in the supplemental materials. This is an interesting robustness finding, and readers should have access to it.

Thank you for the opportunity to revise this manuscript.

(Remarks on code availability)

Reviewer #4

(Remarks to the Author)

Preface: I have conducted this review with a very narrow focus. I have exclusively considered the soundness of the Bayes factor and equivalence test procedures reported. I have also considered 2s remark:

"A smaller technical issue for me was the equivalence tests the authors used. I'd recommend using Bayes factors to test for absence of effects as these don't rely on relatively arbitrary decisions about the smallest effect size of interest."

Finally, I have a remark on the response to the question by reviewer 1:

"Is there a formal test of the (equivalency) claim that the interventions were similar in size to the post-election drop? This is something claimed in the title of the paper, but if there's a formal test of the equivalence in effect size, I missed it."

I defer all other considerations of research quality to the editor and the other reviewers.

Review remarks:

I would like to begin by stating a slight dissatisfaction with Reviewer 2s remark. The reviewer makes it sound like the switch from equivalence tests to Bayes factors is an objective technical correction. It is not. Whether to use Bayesian or frequentist procedures is a matter of epistemological stance, and BFs are in no way strictly superior to TOST for evaluating potential null effects. Bayes factors, like equivalence tests, rely on assumptions that can sometimes be rather arbitrary (e.g. the use of default priors, and what exactly constitutes "strong evidence"). In addition, the common BF approach adopted here gives up on one of the main goals of TOST equivalence testing – to evaluate the practical significance of results. Essentially, the BF herein reported is a Bayesian version of standard NHST. It can only tell you something about how likely it is that the effect is "zero" or "not zero". It is not clear whether a large BF implies a meaningful effect, or if a small BF implies a non-meaningful effect, since the BF is not tied to the SESOI. If you think the SESOI is weak anyway then perhaps this move is appropriate. However, I note that the SESOI was actually based on prior relevant research in this paper. If the SESOI really was considered informative by the authors, it is too bad that it does not seem to be informing the Bayesian analyses at all.

That said, I do not wish to debate whether to use BFs or equivalence tests in the middle of a review process. Both approaches have their strengths and limitations. Speaking as someone who is generally familiar with the Bayesian rationale and BF computation, the procedure implemented here seems completely fine as far as I can see. If the authors want to rely on Bayesian evidence rather than the previously reported frequentist equivalence tests, I don't think it is a very big issue – especially if the SESOI was not very strongly justified. In any case, BFs and TOST results are reported together and along with visualizations of confidence intervals, which gives the reader ample opportunity to decide for themselves which results to rely more heavily on. I think this is a completely fine way to approach the issue. However, I note two problems with the analyses that I think should be addressed:

1. (Minor) The calculation procedure for the BFs should be reported in the main manuscript, not just in the supplementary table text.
2. It seems that BFs are only computed for the non-significant results. This is a common, but inappropriate way to proceed. All results of interest, significant and non-significant, should be subjected to the same BF/equivalence test analyses, and BFs for all results should be reported in the main manuscript. The reason all effects should be evaluated is that...
 - a. (if testing for equivalence with SESOI) there could be results that are statistically significant from zero, but not significantly different from SESOI.
 - b. (if relying on Bayesian evidence) there could be there could be results that are statistically significant from zero, but for which there is not strong Bayesian evidence that the effect is different from zero.

Regarding the question by reviewer 1, the authors reply “Bayes Factor analyses provide moderate to (mostly) strong evidence in favour of the null, as noted in Supplementary Table 5”. I don't see any BFs in Supplementary table 5 that directly corresponds to an effect size comparison between treatment- and time effects. If I have just missed it, feel free to correct me. If not, some formal quantitative assessment of effect similarity between treatment- and time effects should be added to the analyses.

Finally, I find it a bit strange that the analyses and visualizations that directly bear on the title of the paper are relegated to supplementary materials. Unless there is some specific reason for this, I would recommend moving these results into the main manuscript.

I hope the authors find these comments helpful.

Signed: Peder M. Isager (I always sign my reviews)

(Remarks on code availability)

Reviewer 1.

Reviewer #1 (Remarks to the Author):

I enjoyed reading this timely research examining longitudinal data on polarization, meta-perceptions, and corrective interventions in the context of the 2022 Brazilian election. This research represents a substantive methodological and theoretical contribution to our knowledge of (meta)polarization. I applaud the use of preregistered panel data, equivalence testing, and examinations of a theoretically distinct political context. Below I lay out several comments in the hope that they will be useful to the authors as they revise their manuscript.

We are glad you enjoyed the paper and pleased that you appreciate its timely nature, as well as our efforts to tackle the complexities of such an unusual political context, and to pre-register our hypotheses. We are also heartened to read that you consider the research to represent a substantial methodological and theoretical contribution. We hope that you will see just how useful your comments have been in guiding our revisions of the paper.

Perhaps my largest suggestion relates to clarity throughout the manuscript, not only in technical details (methods, analyses), but in also clarifying the contributions. I'll preface by noting that I'm familiar with the difficulty of how to include certain information in the Nature Journal format, where methods come at the end. So some of what I'm looking for is merely a matter of mentioning crucial information earlier, when possible, but some of the information I'm requesting isn't detailed enough anywhere.

This is very helpful feedback and we agree that the original version of the manuscript could have been clearer about several technical aspects and about the contributions of the study. We have adjusted the document in many places with this feedback in mind. In this small part of our item-by-item response we explain these adjustments at a high level, and then will provide further details in our responses to other items of the same review (since most of the points in this paragraph are expanded upon in those items).

To clarify contributions, in summary, we have streamlined the paper around two main points: the lengths we went to in developing measures of affective polarization in the unusual context of Brazil; and the survey experiment, including the rationale behind the particular public policies that we selected for this. To *explain* contributions, our main changes are additions to the results section (where we explain how we went about measuring affective polarization for different political group definitions, with reflections on recent developments of measures for multiparty systems elsewhere), and additions to the discussion section (that outline the theoretical rationale for the chosen treatment conditions).

To clarify methods, we have added a number of paragraphs to the manuscript, and we have improved some important descriptions. Some of these appear in the results section, i.e. earlier in the

paper, as you helpfully suggested. These include more details about the nature of the treatment conditions and the assignment of respondents to different conditions, and a clearer description of the analysis. We have also added clarity to the Methods section and placed several new items in the Supplementary Information.

First, I'd strongly suggest more information about the analyses and modeling choices. There are myriad ways to analyze panel data, ways that are often at odds in terms of what they're trying to isolate, and ways that have long been standard but have now come under strong criticism (e.g., Bell & Jones, 2015; Lucas, 2023). There is a dearth of information in the manuscript about how the analyses are conducted. From what I can glean from the regression tables the authors are using simple Fixed-effects models, but as Bell & Jones (2015) note, these models do not appropriately isolate within-person longitudinal effects. Put another way, I would seem the presented analyses are looking primarily at mean differences between waves or between-subjects differences collapsed across waves. Both of those choices collapse across between-person differences and within-person changes over time (see Curran & Bauer, 2011). As such, I would suggest the author consider alternative modeling strategies that disaggregate within- and between-person effects. Doing so exploits the full value of having longitudinal data, and I think the paper is currently leaving a lot of useful analyses unexamined. Equally important, I would ask that the manuscript provide clear information about the panel-modeling strategies being employed, and how those choices appropriately match the stated hypotheses.

We thank you for these comments about the importance of disaggregating between- and within-subject variability and for providing very helpful references to do so.

We have carefully read the three papers cited in your review, and evaluated our design and analysis approach from their perspective. Their focus is the modelling of time-varying covariates and the challenges that they pose, alongside the inadequacy of fixed-effects models and cross-lagged panel models. We fully agree with their arguments. However, our design is a very specific case to which we believe they do not apply, and realise we should have made this clearer in the original manuscript.

To explain further, the caveats of fixed-effects models that these papers highlight derive from the analysis of observed and non-experimental covariates (e.g. in the test of hypotheses about the effects of engagement in effective coping on stress; effects of negative affect on alcohol or substance abuse or the effects of parental overvaluation affecting childhood narcissism). We fully agree with these arguments - which are critical when predictors are non-experimental and endogenous. In these cases, variation of the predictors between subjects and over time may be correlated with unobserved variables, which may generate important biases in the analysis that are not solved by fixed-effects models.

The specificity of our design derives from the fact that our covariates are exogenous: a) randomly assigned treatment conditions or b) changes over calendar months. These predictors are unrelated

to any specific characteristics of the respondents, rendering omitted variable biases an unthreatening issue. Thus, the exogeneity of our predictors allows for causal inference in a very simple fashion, even with single ANOVA-based comparisons of means as in cross-sectional experiments.

Our use of a fixed effects model - rather than a set of comparisons of means - permits the joint test of hypotheses related to treatment effects implemented in distinct waves, as well as the reduction of error variance with the inclusion of fixed effects. Furthermore, our hypotheses about changes in polarization as related to the proximity to the election consider only variation over time and not the relationship between polarization and time-varying endogenous covariates. Thus, we believe the issue of disentangling between and within-subjects variability (as the predictors related to these two sources are exogenous) becomes trivial enough to be tackled by fixed-effects models.

Nonetheless, to verify this argument empirically, and to make sure we are not missing the point in any way, we have estimated the models suggested by Curran and Bauer, by considering growth curve models rather than fixed effects models and in an alternative specification, adding time as a covariate (as a fixed or random effect). Results remained largely unchanged (see Figs. below).

For the sake of simplicity, we would rather keep the current fixed-effects notation in the main manuscript, including a justification for this choice in the methods section. We are, of course, also happy to replace results with mixed-effects models.

Fig. | Election effects across different model specifications. Panel a depicts the change in average liking scores, compared to the pre-election, during and after the election period for political outgroups. Panels b depicts equivalent effects for perceived intelligence. These effects were estimated using (a) linear regression models with clustered standard errors at the individual level, (b) mixed-effects models with random-intercept at the individual level, (c) mixed-effects models with random-intercept at the individual level and a linear wave trend term, and (d) mixed-effects models with two levels (wave and individual) with a linear wave trend term. All models control for participants' age, sex, household income, education attainment, and region of residence, dummies for wave-specific treatment conditions. Error bars indicate 95% confidence intervals.

Fig. | Treatment effects across different model specifications. Panel a depicts average effects of misperception-correcting information across experimental conditions on liking scores for political outgroups. Panel b depicts equivalent effects for perceived intelligence. These effects were estimated using (a) linear regression models with clustered standard errors at the individual level, (b) mixed-effects models with random-intercept at the individual level, (c) mixed-effects models with random-intercept at the individual level and a linear wave trend term, and (d) mixed-effects models with two levels (wave and individual) with a linear wave trend term. All models control for participants' age, sex, household income, education attainment, and region of residence, dummies for wave-specific treatment conditions. Error bars indicate 95% confidence intervals.

As mentioned, we have also clarified the specification of our models and our empirical strategy in the results section as well as in the supplement. For example, we have added the following to Results:

“We used linear regression models with clustered standard errors at the respondent level to test our hypotheses. We opted for linear models rather than models for bounded variables ease of interpretation of coefficients, and because the dependent variables were not skewed (liking: skewness = .038, $p = .06$, perceived intelligence: skewness = .010, $p = .71$). For hypotheses related to changes in polarization over time, dependent variables were liking/perceived intelligence. Predictors included dummy variables representing mid-election and post-election periods, and a dummy variable indicating if the evaluation refers to an ingroup or outgroup target. Their interactions capture changes in polarization over time. As controls, we included demographic characteristics of the respondent and treatment conditions. The change in the cohort of participants between wave 1 and the panel of waves 2 to 5 precludes the use of participant fixed-effects to assess dynamic changes in polarization (see detailed model specifications in Methods section, and Supplementary Table 3 for participants’ exposure to treatments across waves).

... To assess if these results were affected by differential attrition, whereby the probability of dropping out of the panel is associated with individual level of affective polarization, we performed sensitivity analyses with alternative imputations of the measures for non-respondents; results remained largely unchanged (see Supplementary Fig.4). Moreover, results for trends considering only respondents who were randomly assigned to the control group in our survey experiments are also consistent (as opposed to using the whole sample controlling for effects of the misperception-correcting information treatment) (see Supplementary Fig.5). Results also remain largely unchanged using mixed-effects regression (Supplementary Fig. 6).

... To test randomized treatment effects about affective polarization, we used data from waves 2-5, employing linear regression models and respondent fixed-effects rather than demographic controls to reduce error variance unrelated to the treatment, which is possible given participants were repeated across these waves. Dependent variables were liking / perceived intelligence. The independent variables included a dummy indicating if the target of the evaluation is an ingroup (vs outgroup, varying within subjects), and its interaction with dummy variables representing treatment conditions

(varying between subjects). The coefficient of this interaction term captures treatment effects (i.e. difference in polarization across treatment conditions). We consider only effects of the treatment in the same wave it was dispensed, as our design is not suited to the evaluation of long-term effects. To assess our main treatment effects, we combine data from waves with identical treatment conditions to estimate the average effects of each condition (for example, the assessment of the Amazon condition combines data from waves 2 and 3). We also evaluate treatment effects separately for each wave, see Supplementary Figs. 14 and 15. Results remain largely unchanged using mixed-effects models with random intercepts and trends, as recommended by Curran and Bauer⁴⁰ (Supplementary Fig. 16).”

To Methods, we have also clarified the text in several places.

There are also several details about the interventions themselves that are unclear in the main text, or not discussed at all. One thing that is unclear in the main text is how the intervention is deployed/randomized, for example are people randomized to treatment vs. control at each Wave, or only at Wave 2, from which point that assignment is static? At times it sounds like the effects of the intervention are measured across waves (e.g., intervention at W3, measure at W4), other times it seems like they're at the same wave.

We appreciate the opportunity to clarify this issue that was indeed scarcely developed in the manuscript due to wordcount concerns. We have explained this more fully now in Results and Methods, and, for full transparency and clarity, added to the Supplement: i) translations of full text of each of the treatment conditions (Supplementary table 1), and ii) a table showing the flow of all respondents to different control/treatment conditions for waves 2-5 (Supplementary table 3).

Results now includes the following, explaining how assignment to control/treatment conditions worked for each wave:

“Starting with wave 2, we randomly assigned respondents to control (not receiving misperception-correcting information before evaluating liking and intelligence of groups) or to receiving one of two treatment conditions of misperception-correcting information prior to encountering feeling thermometer questions. Our treatment conditions first provided respondents with a neutral briefing about a public policy, stating that some people support it while others do not (see Supplementary Table 1 for text of treatment conditions). Respondents assigned to a treatment condition were then asked to estimate how many lulistas and bolsonaristas out of 10, on average, support that policy. They were

subsequently informed about those groups' actual level of support, using our data on policy support from wave 1, to reveal the gap between their own estimations and the earlier findings of the same survey that they were taking part in.

In each wave, we assessed the effects of misperception-correcting information relating to two, different controversial policies. In waves 2 and 3, we used as treatment conditions: (i) a policy of legalization of abortion up to the third month of pregnancy, and (ii) a policy of zero deforestation in the Amazon. Respondents were initially randomly assigned to control or to either treatment condition, with those assigned to control in wave 2 also assigned to control in wave 3. Those assigned to a treatment condition in wave 2 received the alternative condition in wave 3. In waves 4 and 5, we assessed treatment conditions about affirmative action policies that assist those from low-income households, and Afro-Brazilians and indigenous people in gaining access to higher education³⁹: (iii) social quotas, and (iv) racial quotas, respectively. The same approach to assignment was repeated as per waves 2 and 3. In wave 4, respondents were randomized independently of their assignments in prior waves. Then those assigned to control remained there across waves 4 and 5, and those assigned to a treatment condition in wave 4 were counterbalanced in wave 5. Supplementary Table 3 summarizes the flow of all panel respondents through assignment to control or experimental conditions for all survey waves.”

The reader is then reminded of this design in Methods, with respect to the analysis:

“To test hypotheses about misperception-correcting treatments effects on affective polarization, a 2 (evaluated group: ingroup vs outgroup) x 3 (content: control vs treatment 1 vs treatment 2) design was used. To test hypotheses about effects on issue polarization, a 2 (order: before vs after treatment, within subjects) x 2 (political self-identification: lulista vs bolsonarista, between subjects) x 2 (policy: treatment 1 vs treatment 2, between subjects) mixed design was used, with replicates for other political identification groupings (left vs right, and anti-lulista vs anti-bolsonarista). To assess our main treatment effects, we combine data from waves with identical treatment conditions to estimate the average effects of each condition (rather than performing simple comparisons of means over separate waves). For example, the assessment of the Amazon condition combines data from waves 2 and 3.

... A regression model similar to the one used to test treatment effects, with additional interaction terms with a dummy variable representing the wave, was used to evaluate differences in treatment effects across waves (i.e. the difference between wave 1 and 2 for abortion/Amazon treatments, and the difference between wave 3 and 4 for social/racial quotas treatments).”

It’s also unclear whom participants are getting corrective information about, at points it sounded like participants are getting information about ingroup attitudes, outgroup attitudes, and/or general population attitudes. This all needs to be clearer in the main text of the manuscript.

This is a very important point and in response we have added a number of clarifying edits at several points in the manuscript.

Indeed, participants always receive corrective information about both lulistas and bolsonaristas and hence, about ingroup and outgroup attitudes (when the respondents themselves provide a political affiliation, that allows for the identification of ingroup/outgroup classification). As noted above, we have now added to the Supplementary Information full script of the four treatment conditions. For example, the supplement item showing the abortion condition (Supplementary table 1) shows the following:

Abortion Condition:

Currently, abortion is permitted in Brazil only in three situations: when there is a risk to the woman's life, when the pregnancy is a result of rape, or when the fetus is anencephalic (has not developed part or all of the brain). In all other cases, abortion is prohibited in Brazil.

However, there is significant debate surrounding abortion. Some people advocate for relaxing the rules to allow abortion for any reason, but under the guidance of professionals and provided it is done within the first three months of pregnancy. Others oppose this proposal, often arguing for even stricter regulations on abortion.

Now, think about Bolsonaristas, those who intend to vote for Jair Bolsonaro in the first round of this year's elections. Out of every 10 Bolsonaristas, how many do you believe support the proposal to relax abortion regulations to allow it up to the 3rd month of pregnancy? __

Now, think about Lulistas, those who intend to vote for Lula in the first round of this year's elections. Out of every 10 Lulistas, how many do you believe support the proposal to relax abortion regulations to allow it up to the 3rd month of pregnancy? __

In an earlier round of this study, we asked Lula supporters and Bolsonaro supporters their opinions on the decriminalization of abortion up to the 3rd month of pregnancy.

Very few Bolsonaro supporters, only 11%, said they were in favor of decriminalizing abortion up to the 3rd month of pregnancy. The percentage of supporters was slightly higher among Lula supporters. However, even so, less than half (only 46%) of Lula supporters indicated that they were in favor of decriminalizing abortion.

To the Results section, we have also clarified where this information came from, and why we chose to use lulista/bolsonarista responses:

“We also obtained data in the first wave about policy opinions on controversial issues that we then used to construct experimental treatments. These treatments (see Supplementary Table 1 for details) drew on responses to policy-support questions from lulistas and bolsonaristas rather than other political group definitions because this group definition elicited the largest affective polarization scores. The second wave recruited a different sample of participants, who were then repeatedly contacted over the subsequent waves, creating a panel structure (see details of sampling in the Methods section and descriptive analysis of the sample over waves in Supplementary Table 2).”

The source of the information is also noted in a previously quoted paragraph:

“They were subsequently informed about those groups’ actual level of support, using our data on policy support from wave 1, to reveal the gap between their own estimations and the earlier findings of the same survey that they were taking part in.”

I would also suggest the paper do more to spell out the theoretical and methodological contributions of examining polarizations and meta-perceptions in such an atypical political system/culture. What are the implications for intergroup theory precisely (e.g., how do the findings challenge past theorizing that is built on data from only US/European political contexts), and what are the methodological implications? On this latter point I think the manuscript as written undersells itself. I found their measurement and operationalization(s) of identity/political boundaries quite novel, but one only realizes this after reading the methods section. While reading the main text I had a reaction of “well you keep telling me this is a political context that is theoretically distinct from places like the US, but you’re still using a binary in/out-group boundary operationalization like researchers in the US do.” It was only upon reading the methods that I truly appreciated the immense thought the authors put into how they operationalized these identities, and how what they did makes a significant theoretical and methodological contribution. I would suggest that authors better foreground these contributions in the manuscript! Personally, if I were someone considering how to study polarization in a political context unlike the US, I’d want to use the methods these authors used here!

This is an excellent and well-received series of related points, and we were rather light on this aspect previously due primarily to word-count concerns. We have added what we believe to be a series of improvements in different parts of the document, placing a much fuller explanation of our methodological choices to assess polarizations in Results (to avoid the reader not meeting it until Methods).

First, in the abstract, we’ve noted the different groups that we consider:

“Given the complex nature of Brazil’s political system, we document horizontal affective polarization of combinations of political ingroups and outgroups, including left-right position-holders, negative partisans, and supporters of the most important presidential candidates.”

To the Results, we have explained what we did and why we did it in much more detail, and done so by including some comparative points about our approach in relation to recent developments in the measurement of affective polarization in multiparty (primarily European) systems. We also emphasize here the important point that we specifically measure *horizontal* polarization, which we also undersold in the original draft of the manuscript:

“We measured perceived intelligence as well as liking, as the former is regarded as a coarser instrument to capture more extreme sentiments that stray towards outgroup dehumanization, and may also capture an aspect of distrust in the form of perceived competence. Respondents were asked to score sentiment towards groups of citizens of various political persuasions (i.e. “horizontal”³³ or “social”³⁴ polarization), providing assessments of different camps of distinct definitional types of sociopolitical groups: left-right position-holders, lulistas and bolsonaristas, anti-lulistas and anti-bolsonaristas, and “petistas” and “anti-petistas” (i.e. supporters and non-supporters of Brazil’s only party to enjoy significant levels of voter identification, the Workers’ Party²⁸). The first survey wave enabled us to compare frequencies of respondents’ ingroup self-categorizations, as well as the frequencies of co-occurrences, of different self-categorizations (see Supplementary Fig. 1 for details of co-occurrences, including two varieties of pro-politician categorization, “being comfortable calling oneself a lulista / bolsonarista” and intention-to-vote for Lula or Bolsonaro in the first-round elections, which is the categorization used throughout the Results).

Beyond the question of which groups to measure, the Brazilian context presents additional complexities. For example, recent developments in the assessment of affective polarization in multiparty systems weight liking scores using party vote share^{14,15,34}. These are not straightforwardly applicable in the Brazilian case since the huge number of politically relevant parties means that thorough liking batteries would be impractically long. Politicians’ frequent party switching also makes this a problematic approach to weight their (non-)supporter groups, as per Reiljan et al³⁴, because it implies weak (and variable) symbolic correspondence between politician and party (although other measures of electoral relevance could be used). Indeed, Bolsonaro, was not a member of any party for a portion of his presidency, and has represented nine different parties during this career. Another issue is the non-inclusion in aggregate measures of affective polarization of respondents who do not identify with an ingroup, and thus inevitably have no individual ingroup-outgroup feeling-thermometer gap. Some scholars minimize this problem using definitions such as “leaning” towards one political party over others (i.e. survey respondents feeling “not close” but “a little closer to”)¹⁴; others instead do not require respondents to express even weak support, and define ingroup as the most-liked target (e.g. party), among multiple assessed targets of the same type (i.e. other parties)¹⁵. For simplicity and consistency across sociopolitical group types, we considered single (rather than composite) outgroups, and did not apply weighting. Hence, pro-politician outgroups were considered to be supporters of the

one other important presidential candidate (i.e. lulistas vs. bolsonarista), although alternative approaches are possible using our data (i.e. lulistas vs. anti-lulistas). In our first wave, using intention-to-vote for Lula or Bolsonaro in first-round elections to define lulista and bolsonarista groupness, excluded 36.82% of respondents. This is a similar percentage to those Reijlan¹⁴ reports for his affective polarization index applied to European countries. Using ideological self-placement (left-wing and right-wing) to define groups excluded 43.87% of respondents, and using our anti-politician definition excluded 22.38%. Of note, given recent arguments about the depth and persistence of the petistas vs. anti-petistas division in Brazilian politics²², our first-wave feeling thermometer assessment of excluded 51.13% of respondents when the petistas vs. anti-petistas grouping was used, and had lower affective polarization scores (liking: M = 46.87, intelligence: M = 41.57) than lulistas vs. bolsonaristas (liking: M = 59.20, intelligence: M = 53.85).”

The addition of a new supplementary figure, referred to at the end of the first of the paragraphs above, should also assist readers in making sense of the different political affiliation categories in the study:

Supplementary Fig.1 | Co-occurrence of political affiliation considering alternative

grouping criteria. The figure shows the number and percentage of respondents who identify themselves as part of different groups concomitantly. For example, 568 respondents identify as both lulistas and left-wing. In percentage terms, this corresponds to 76% of lulistas identifying as left-wing. lulista and bolsonarista are defined according to intention to vote for the respective candidate in the first round of the presidential election as reported in wave 2. Left and Right are defined according to self-reported identification on the political spectrum as reported in wave 2. Anti-Bolsonaro, anti-Lula, anti-neither, and anti-both are defined according to participants' indication that they would never vote for Bolsonaro, Lula, neither of these candidates, or either of these candidates, respectively, as reported in wave 2. And Self-Lula and Self-Bolsonaro refer to self-reported identification as a lulista or a bolsonarista.

Minor Comments:

- The preregistrations state discrete, numbered hypotheses. I would suggest referring to these numbered hypotheses in the manuscript.

Thank you again for this suggestion. We agree this would help the reader to follow the paper. Given that we have many hypotheses (indeed, 12 pre-registered hypotheses are tested in the paper, all referring to dynamics of polarization or to treatment effects), we were also mindful of clogging up the earlier parts of the manuscript in such a way as to reduce readability. As a compromise between these competing pressures, we decided to summarise the two families of pre-registered hypotheses in the introduction, in line with having reframed the paper more around our main contributions.

Hence, the introduction now includes the following sentence:

“Our pre-registered hypotheses anticipated that: i) the moment of the elections would heighten affective polarization, and ii) that experimental treatments correcting misperceptions of group positions on controversial issues would reduce it (see Methods for pre-registration details).”

- Is there a formal test of the (equivalency) claim that the interventions were similar in size to the post-election drop? This is something claimed in the title of the paper, but if there's a formal test of the equivalence in effect size, I missed it.

This is also a helpful point, although we should point out that our title claim proposes only that these effect sizes are “similar” rather than identical.

We have added Supplementary Fig. 17 (added below) which allows for a visual comparison of standardized coefficients representing changes over time (election and post-election as compared to

pre-election) and treatment effects. The figure shows the similarity of the effects and the overlap of the confidence intervals.

We also conducted some analyses. These do not reject the null hypotheses of equality of coefficients and Bayes Factor analyses provide moderate to (mostly) strong evidence in favour of the null, as noted in Supplementary Table 5.

Supplementary Fig. 17 | Election and treatment effects. Panel a depicts the standardised change in average liking scores, compared to the pre-election, during and after the election period for political outgroups (blue coefficients) and average standardised effects of misperception-correcting information across experimental conditions on liking scores for political outgroups (gray coefficients). Panels b depicts equivalent effects for perceived intelligence. These effects were estimated using linear regression models with clustered standard errors at the individual level. Models for the election effects control for participants' age, sex, household income, education attainment, and region of residence, dummies for wave-specific treatment conditions. Error bars indicate 95% confidence intervals.

I hope the author(s) find my comments helpful as they revise their manuscript.

Thank you! Very much so.

Cited:

Bell, A., & Jones, K. (2015). Explaining fixed effects: Random effects modeling of time-series cross-sectional and panel data. *Political Science Research and Methods*, 3(1), 133–153. <https://doi.org/10.1017/psrm.2014.7>

Curran, P. J., & Bauer, D. J. (2011). The disaggregation of within-person and between-person effects in longitudinal models of change. *Annual Review of Psychology*, 62(1), 583–619. <https://doi.org/10.1146/annurev.psych.093008.100356>

Lucas, R. E. (2023). Why the cross-lagged panel model is almost never the right choice. *Advances in Methods and Practices in Psychological Science*, 6(1), 25152459231158378. <https://doi.org/10.1177/25152459231158378>

Reviewer #1 (Remarks on code availability):

The code is Stata code, and I do not have Stata nor do I know how to use it. I use R exclusively.

We have now updated the code.

Reviewer 2.

Reviewer #2 (Remarks to the Author):

In the paper "Sociopolitical Polarization Reductions After Elections Are Similar in Extent to Correcting Misperceptions about Controversial Policies" the authors investigate the dynamics of affective polarization before and after the Brazilian election and whether providing information on the beliefs of the outgroup can reduce polarization. While the paper provides some interesting results, I don't think that the results are sufficiently significant to warrant publication in a high-impact journal like Nature Communications. The main take-aways from the paper are that there is high affective polarization between Bolsonaro supporters and Lula supporters, that this polarization decreased somewhat after the election and after correcting misperceptions about the beliefs of the outgroup, and that the FIFA world cup had not meaningful effects on polarization. None of these effects are very surprising and although they provide an addition to the literature, they are not very novel. The most interesting effect in my opinion is that polarization decreases after providing information about the outgroups' beliefs about controversial issues. While certainly interesting, this effect is only an extension of earlier work (1), as the authors themselves note.

We thank you for these comments, and for noting that our results are interesting. We also thank you for the nudge to articulate and emphasize our contributions more fully and clearly, especially those

that have been highlighted as substantive contributions by other reviewers. The updates to the manuscript do this, in particular, taking into account your very helpful suggestions around framing.

Hence, the paper is now more tightly framed around: (a) how we went about measuring and assessing affective polarization in such an unusual political context (and the reasons for our methodological choices); and (b) the part that you consider the most interesting: the change in outgroup animosity following correcting misperceptions of groups' views on controversial issues.

Point (a), regarding measurement choices, is explained, and the relevant additions to the paper are quoted at length in response to Reviewer 1's comments above. (In a nutshell, Reviewer 1 considers the affective polarization measurements used in this unique political context as important contributions to highlight, which we have now done throughout the manuscript.) Since you do not emphasize this aspect, we will not repeat the quoted sections here.

Point (b) is a very important part of the paper – thank you for encouraging us to better emphasize it - and one that we have explained with greater reference to theory. To better emphasize this aspect of our contribution, we have added high up in the manuscript:

In the Abstract:

“Experimental treatment conditions correcting misperceptions of group stances on single, controversial policies reduced affective polarization by up to 6.66 percentage points (liking thermometer), and, in some instances, also shifted policy support. We discuss these findings in light of attitude moralization, accessibility and persuasion knowledge24/11/2024 08:46:00.”

And in the introduction:

“And while various misperception-correction treatments to reduce affective polarization have been tested sufficiently frequently in the United States to warrant meta-assessment in two recent papers^{20,21}, they have been far less commonly trialled in other countries, and, in our view, their varied effects warrant further reflection.

... The experimental treatment conditions that we selected to correct misperceptions were deliberately focused on controversial issues, in the sense that they are morally charged topics about which opposing opinions are commonplace, and they feature with some regularity in public discussion. While views on abortion in the first three months of pregnancy, for example, may be entrenched, the moral character of this issue also implies that misperception corrections may be highly informative³⁰, and the topic's frequent featuring in public debate suggests the possibility of increased accessibility^{31,32}. The effect

sizes that we find are modest with respect to some other varieties of intervention to reduce affective polarization, however, they are reasonably substantial in comparison to other policy-stance misperception corrections. We suggest there are normative benefits to identifying effective treatments of this type, given their focus on substantive, programmatic content, as well as practical benefits of brevity, low-cost and scalability.”

A richer explanation of why we developed treatments around controversial issues is now included in Discussion. In short, theory suggests that such topics may lead to somewhat larger effects than non-controversial issues, and there are normative and practical arguments as to why short, policy-stance misperception corrections may be worthy of further investigation (especially outside of the US context and during moments of heightened political tension):

“We deliberately considered controversial policy issues, in light of indications in the literature that they may be particularly useful for assuaging outgroup animosity. For example, Ryan⁵⁸ draws on the psychology literature, to argue that moral attitudes, especially those for which views are deeply-held and opposing, engage “a distinctive mode of processing” because they “arouse certain negative emotions”. Similarly, moral reframing is known to be highly effective at shifting support for and against political candidates⁵⁹. Taking the argument further, most definitions of “controversy” entail not only opposing views that tap into moralized issues, but also an element of commonplace interest or widespread discussion⁶⁰. If a topic is particularly prominent, then some increased accessibility is implied, suggesting enhanced cognitive processing of messages relating to these issues^{31,32}. Our most effective treatment condition, policy stance on abortion rights in the first three months of pregnancy, like the other conditions explored here, appeals to moral foundations additional to that of loyalty/betrayal^{61,62}. Indeed, abortion is often treated as an archetypal moral issue in the literature; in the United States, it is frequently used to study the antecedents⁶³⁻⁶⁵ and consequences⁶⁶ of attitude moralization (the extent to which people perceive attitudes to be imbued with moral convictions). In Brazil, moral values have been demonstrated to underpin physicians’ willingness to work in the area of abortion, even in circumstances where it is straightforwardly legal⁶⁷. When we then turn to consider commonplace interest or widespread discussion, an easily available indicator is the relative frequency of Google searches. Upon restricting search data to Brazil, we find that abortion has also been searched for consistently more often than the other policy issues used in our treatment conditions (see

Supplementary Fig. 26). Our second-most effective treatment, Amazon deforestation, is the second-most commonly searched for topic.

Changing people's views towards sociopolitical groups or towards policies using misperception correction involves providing an external signal strong enough to overcome the internal psychological costs of change. On the one hand, controversial issues are associated with entrenchment of views, as established in the persuasion knowledge literature⁶⁸, implying high costs of changing one's attitudes. However, on the other hand, that same literature argues that group stances on controversial issues are also likely to be strongly diagnostic, and thus highly informative³⁰. Thus, the patterns of effect sizes that we find for our i) affective polarization and ii) issue polarization outcomes may reflect the interplay between the variable signal strengths of the different misperception-correction conditions, and respondents' individual psychological costs of i) adjusting their degree of like (or dislike) and perceptions of intelligence towards sociopolitical groups, and ii) of revising their stances on policies. Respondents may have especially entrenched views on the issue of abortion, for example, but learning that they wildly mischaracterize outgroup opinion on this issue may especially strongly prompt re-assessment of outgroup likability. This suggestion is an area for future research, which we encourage for policy misperception corrections. Although these may not generally be as effective at reducing partisan animosity as some other approaches, they have a normative advantage beyond the practical considerations of cost and scalability: for those who wish to encourage programmatic politics, they shift voters' attention to substantive policy issues, as opposed to groups' demographic differences, for example⁴²."

The reframing outlined above means that we have substantially reduced the part about the FIFA World Cup and the 8 January riots. Following guidance from the editor, we have not completely removed the World Cup and 8 January material from the main manuscript.

On the positive side, the study reported in the paper as well as the analyses seem very sound and are well-reported.

Thank you for noting this.

A smaller technical issue for me was the equivalence tests the authors used. I'd recommend using Bayes factors to test for absence of effects as these don't rely on relatively arbitrary decisions about the smallest effect size of interest.

We appreciate this suggestion, and thank you for it. We have estimated Bayesian models and added a table with Bayes factors to the supplement (Supplementary Table 5), now referring to the Bayes factor results throughout the manuscript, in place of equivalence tests).

We are pleased to be able to report that the null effects referred to in the original version of the manuscript are either strongly or very strongly supported with Bayes factors (as well as by the equivalence tests that we originally submitted), in almost all cases. The only exceptions to strong or very strong Bayes-factor support are always moderately supported, and are limited to some of the World Cup analyses (which are strongly de-emphasized in the revised manuscript, informed by these results) and to one part of a new analysis we have added to the Supplement response to a suggestion from Reviewer 1. This is an assessment comparing the post-election decline in affective polarization to the treatment effects (note that this assessment was motivated by the title claim, however our only claim is of "similarity" rather than identicalness between the post-election decline and the effect sizes of multiple different treatment conditions). When conducting this analysis, we found that a quotas treatment was only moderately supported.

On the matter of the equivalence tests, we decided to mention them in the Supplement in a table alongside all of the Bayes factors calculated. This is because when the findings of the two approaches converge, as ours do, the robustness of the findings is strengthened (Lakens et al, 2020, Lakens, 2020). While we agree the magnitude of the SESOI is not grounded on an undisputed basis, equivalence tests have a similar goal to Bayes factors. The primary differences between the equivalence tests and Bayes factors are philosophical (bayesian vs classical probability approaches) and practical (how to incorporate prior information to statistical inference). Due to these differences, results are not necessarily convergent across approaches. Hence our preference to include both sets of tests in the Supplement, though we are happy to exclude them upon request.

Lakens, Daniël, Anne M. Scheel, and Peder M. Isager. "Equivalence testing for psychological research: A tutorial." *Advances in methods and practices in psychological science* 1.2 (2018): 259-269.

Lakens, Daniël, et al. "Improving inferences about null effects with Bayes factors and equivalence tests." *The Journals of Gerontology: Series B* 75.1 (2020): 45-57.

Another recommendation would be to drop the analyses on the impact of the FIFA world cup on polarization. In my view, the rationale behind expecting effects of the event on polarization is rather weak and the results are also not very noteworthy. I think dropping this analysis would improve the readability and focus of the paper.

Thank you for this suggestion, and for the overall comments about reframing the paper. As noted above, we have reduced the part about the World Cup to a brief summary at the end of the part of the Results that talks about dynamics, but not removed it entirely on the advice of the editor.

References

(1) Voelkel, J. G. et al. Interventions reducing affective polarization do not necessarily improve anti-democratic attitudes. *Nat Hum Behav* 7, 55–64 (2022).

Reviewer #2 (Remarks on code availability):

I could not access the code, because I needed permission.

We have uploaded an updated version of the data and code, making sure it is publicly available in the following link: https://osf.io/y8xkm/?view_only=473000fa5ea64be293f7d7ad977f34b9

Reviewer 3.

Reviewer #3 (Remarks to the Author):

This paper assesses changes in the levels of affective polarization in Brazil in the context of the last presidential election in this country. The papers use a four-panel wave conducted before, after, and during the election. This data is augmented with experimental instruments embedded in the surveys to measure the effects of correcting misperceptions of ingroup and outgroup options on affective polarization. The paper is well-written, well-motivated, and carefully developed. With that said, I have suggestions to offer in the structure of the paper and the methodological approach.

Thank you for noting that the paper is well-motivated, and carefully developed.

Framing

In the structure of the paper, in my view, the descriptive content about the stability of the levels of polarization, and the null effects of the world cup and the January 8th riots are the least exciting components of the paper. As a matter of fact, in my view, the authors spend too much time discussing these dynamics, when they merely serve the purpose of showing that polarization in Brazil is stable and hard to change. This is the main message of the paper's introductory section, which can be presented in a single section instead of in three results sections.

The decision to have the World Cup and the riots as separate sections of the paper distracts the reader from the most exciting results of the paper: the effects of correcting misperceptions on levels of polarization. Even though the authors bring novel data, the

fact that Brazil is a polarized country has been documented already (as some of the literature cited by the authors show, Zucco and Samuels' work and Ortelado, and more recently work by Nunes).

...So I suggest the authors move these two sections to the end of the paper or incorporate them as smaller paragraphs in the first results section.

This is very helpful advice. As noted in our response to the other reviewers, the paper is now more tightly framed around: a) how we went about assessing affective polarization in such an unusual political context (and the reasons for our methodological choices); and b) the change in affective polarization following correcting misperceptions of groups' views on controversial issues.

We have also followed your specific and helpful advice to much-reduce the World Cup and riots in Results, and reorganise Results somewhat, removing that part as a sub-section. There is now a brief summary of the World Cup and riots at the end of the part of Results that discusses trends in affective polarization over time. We thank you for this suggestion! Following a consultation with the editor, we decided to move much of the previous material to the Supplement, along with the corresponding figure (what was Fig. 2).

However, serious policy evaluations of interventions to reduce are still more scarce, and I think this is where the paper's main contribution resides.

This is an important point. As noted in the response to Reviewer 2 above, we have mentioned this in the Introduction. We also add to the Discussion:

“We also encourage other researchers to contribute to knowledge about the effectiveness of experimental treatments to reduce affective polarization outside of the United States context.”

At the same time, with this reorganization, the paper will become more about measuring interventions to reduce polarization. So the authors will need to give higher centrality to this topic in the introduction of the paper. This is already there, but the authors must flesh out a bit more.

We thank you for this point, too. Indeed, the paper has now been substantially reoriented as suggested here, with new arguments and sections of text added about both measurement and the rationale of our interventions. (The new text about measurement is quoted in response to Reviewer 1's helpful suggestions, and the next text about the rationale of our interventions has been quoted in response to Reviewer 2's, so I will not requote them here - but very much hope that you consider those additions.)

As you suggested, the Introduction has also been adjusted to make clear the greater centrality of measuring interventions to reduce polarization:

“Drawing on data from five unique and nationally representative survey waves, with panel responses composing the final four, we employ a diverse set of ingroup-outgroup assessments to follow dynamic variations in affective polarization among groups of citizens throughout the 2022 electoral period, and into 2023. We also assess the effectiveness of four different, single-issue experimental treatments to correct misperceptions of ingroup and outgroup stances towards controversial public policies. Our pre-registered hypotheses anticipated that: i) the moment of the elections would heighten affective polarization, and ii) that experimental treatments correcting misperceptions of group positions on controversial issues would reduce it (see Methods for pre-registration details).”

We also now open the Abstract with:

“Affective polarization is little-studied in systems that present ambiguity in relevant political groups.”

Methods

Regarding the methodological aspects, I have some questions about the models used in the experimental section. The authors mention their treatment is an intervention to correct misperceptions. So on average, voters believe X people to support policy Y, but the actual number is that on X is smaller than what the average voter believes. While on average, the treatment is correcting misperceptions, the treatment is actually doing different things conditional on where the voter i is in the full distribution.

In the current form, the authors present the results by treating their intervention as an intention-to-treat effect. This is fine, and I think the authors should keep this available for readers as the primary analysis. However, the authors have much richer data, and I would like the author to provide extra analysis that could be incorporated into the paper using a few different options to measure the treatment effect.

First, I would like the authors to present these results separating by subgroup between those in which the treatment did exactly what the authors designed for theoretically (reduce misperceptions), and those who already had a view close to or smaller than the actual number.

This can be done by splitting the data, or the authors could estimate the Complier Average Causal Effect using a IV setup. Splitting the data seems actually a superior choice because this is not a matter of compliance, but more of the treatment potentially be pushing respondents in different directions.

Lastly, I would also be interested in some type of continuous measure of the correction. For example, I imagine the effects are much larger, given how wrong I was about an outgroup. This is important to be added to the paper.

We thank you for considering our results so carefully, and for these suggestions. When we were originally conceiving the survey experiment, we thought hard about whether to design it so that we would be able to make sense of how the size of respondents' misperceptions might relate to the treatment effect size – the issue at the core of this request. In the end, we decided not to go very far down this route for primarily theoretical reasons, which set up what we judged to be insurmountable practical challenges.

Theoretically, it did not seem reasonable, given different arguments in the literature (especially the persuasion knowledge literature), to straightforwardly hypothesize the intuitive relationship: that the greater the size of misperception that our survey experiment illuminates to the respondent, the greater the reduction in the respondent's affective polarization. Rather, for example, very large misperceptions might elicit greater scepticism towards the veracity of the correction information that our treatments provide, which could lead to smaller rather than to larger treatment effects. This is suggested from work on believability (eg Cone et al 2019) and on counter-attitudinal persuasive messaging by Tormala & Petty 2004 (they find that sources of "expertise" - as our survey results might be considered by the respondents - make original attitudes more concrete, rather than shift them, when people are minded to resist persuasion). While we did consider questions to probe how believable respondents took our misperception corrections to be, we were concerned that these would be insufficiently reliable because they were particularly likely to elicit social desirability bias.

Citations:

Cone, Jeremy, Kathryn Flaharty, and Melissa J. Ferguson. 'Believability of Evidence Matters for Correcting Social Impressions'. *Proceedings of the National Academy of Sciences* 116, no. 20 (14 May 2019): 9802–7. <https://doi.org/10.1073/pnas.1903222116>.

Tormala, Zakary L., and Richard E. Petty. 'Source Credibility and Attitude Certainty: A Metacognitive Analysis of Resistance to Persuasion'. *Journal of Consumer Psychology* 14, no. 4 (January 2004): 427–42. https://doi.org/10.1207/s15327663jcp1404_11.

Given the multiple possible predictions emerging from the literature, we realised that we would need a large sample size to properly investigate this issue using subgroup analyses. Hence, practically speaking, from having previously run online surveys in Brazil, we realised that sustaining the required sample size would be extremely difficult in a panel, if we were also aiming for a reasonable degree of population representativeness.

To explain a little more about the power requirements to investigate subgroups here, and hence why a much larger sample size would be needed, our treatment effects are estimated using 2-way interactions between the treatment condition and an indicator variable that captures if the respondent is considering an ingroup or outgroup target. Thus, we can certainly evaluate differences in treatment effects across subgroups. However, to move beyond simple point estimates and evaluate whether subgroup treatment effects are significantly different from one another, we need

to run 3-way interactions (a test of difference between subgroups' effects is logically equivalent to this).

Given the above arguments, and with the desire to respond as fully as possible to your request, we have conducted many different analyses that consider multiple ways of thinking about the extent of misrepresentation (e.g. relative to a respondent's initial estimate of outgroup opinion, relative to respondent's initial estimate of ingroup-outgroup gap in opinion, and so on):

- **Operationalization misperception 1.** This measure is operationalized as the difference between the estimated and the actual percentage of lulistas who support abortion (46%) or the difference between the actual (63%) and estimated percentage of bolsonaristas who support zero deforestation in the Amazon. Negative values--i.e., misperceptions that are on the opposite direction of the expected--are turned into zero.
- **Operationalization misperception 2.** This measure is a self-reported feeling of surprise upon receiving the information that (i) a minority of lulistas support abortion or (ii) a majority of bolsonaristas support zero deforestation. It is operationalized as a dummy that equals 0 if the participant reports no surprise at all and 1 if the participant reports being a bit, moderately, or very surprised.
- **Operationalization misperception 3.** This is a categorical variable that classifies if participants have (i) stereotype-consistent misperceptions (i.e., if they indicate that between 60 and 100% of lulistas support abortion or if they indicate that between 0% and 40% of bolsonaristas support zero deforestation; in fact, 46% of lulistas and 63% of bolsonaristas support the respective policies), (ii) stereotype-inconsistent misperceptions (i.e., if they indicate that between 0% and 30% lulistas support abortion or that between 80% and 100% bolsonaristas support zero deforestation), or (iii) accurate perceptions (i.e., if they indicate that between 40% and 50% lulistas support abortion or between 50% and 70% bolsonaristas support zero deforestation).
- **Operationalization misperception 4.** This is a dummy variable that indicates whether or not people have inaccurate perceptions about outgroup support for the policy. We first calculate the difference between perceived outgroup support for the policy and the actual figures (abortion: 46% lulistas and 11% bolsonaristas; zero deforestation: 80% lulistas and 63% bolsonaristas). We consider underestimation of outgroup support if this difference is lower than -1, accurate perceptions if it is between -1 and +1, and overestimation if it is greater than +1. We then construct the misperception 4 dummy such that people are inaccurate if they underestimate outgroup support for zero deforestation (irrespective of the participants' group) or if bolsonaristas overestimate the support for abortion among lulistas or if lulistas underestimate support among bolsonaristas.

We have estimated models with both outcomes of affective polarization (in terms of liking and intelligence) across different measures of misrepresentation. We found mostly null results and none after using the Benjamini-Hochberg correction for multiple comparisons.

Fig. | Adjusted critical P -values for the 3-way interactions interacting degree of misperception, treatment condition, and the outgroup dummy via the Benjamini-Hochberg correction. Panel a depicts the p -values for the 3-way interaction interacting degree of misperception, treatment condition, and the outgroup dummy. To account for multiple hypothesis testing and control for the false discovery rate in multiple comparisons, we adjust critical P -values via the Benjamini-Hochberg correction. The adjusted p -values are presented in panel b. The red line indicates the 5% significance level.

Furthermore, considering these four measures of the extent of misrepresentation, we found less than 30% of the interactions with strong support for the null as estimated by Bayes factor analyses. Hence, we are neither able to consistently detect heterogeneous subgroup effects, nor to rule them out by means of Bayes factor analysis. We agree that this is a shortcoming of our paper, however, it was inevitable given the practical considerations of sustaining the panel.

Fig. | Bayes factor (BF₁₀) calculations for the 3-way interaction interacting degree of misperception, treatment condition, and the outgroup dummy. This figure depicts the BF₁₀ values for the three-way interaction effect across different measures of misperceptions and treatment conditions. Values below the orange line indicate strong evidence for the null hypothesis (i.e., absence of an effect), whereas values above the red line indicate strong evidence for the alternative hypothesis (i.e., presence of an effect). The higher the BF₁₀ value, the more evidence one has for the alternative hypothesis.

We have also conducted post-hoc power assessments (using, in the absence of clear ex-ante expectations of effect sizes, the magnitude of the effect sizes of the main effects observed in our models). These calculations found that we do not have sufficient power to pursue three-way interactions – as anticipated given our pre-survey discussions on this issue. Indeed, they indicated that our sample size provides very low statistical power for such assessments: below 50% in all but one case (see figure below).

Fig. | Statistical power calculations for the 3-way interaction interacting degree of misperception, treatment condition, and the outgroup dummy. This figure depicts the statistical power across different measures of misperceptions and treatment conditions, as determined by changes in R-squared values. The statistical power was computed using the R package InteractionPowerR, considering the following parameters: (a) sample size, (b) regression coefficient of the 3-way interaction term (misperception measure \times experimental condition \times outgroup dummy), (c) Pearson's correlation between the misperception measure and liking scores, (d) Pearson's correlation between the experimental condition and liking scores, (e) Pearson's correlation between the outgroup dummy and liking scores, (f) Pearson's correlation between the interaction term (misperception measure \times experimental condition) and liking scores, (g) Pearson's correlation between the interaction term (misperception measure \times outgroup dummy) and liking scores, (h) Pearson's correlation between the interaction term (experimental condition \times outgroup dummy) and liking scores, (i) Pearson's correlation between the misperception measure and the experimental condition, (j) Pearson's correlation between the misperception measure and the outgroup dummy, and (k) Pearson's correlation between the experimental condition and the outgroup dummy.

Still in the methods section, these different analyses could also allow the authors to speak to the issue of social desirability bias. It is hard to disentangle if the experimental results come from the fact that respondents are adjusting their polarization levels because they hold more accurate views about the outgroup, or because the survey is just simply correcting them. Looking at the sensitivity of the results conditional on the gap and the subgroups could also allow the authors to speak to this issue.

We agree that social desirability bias may be playing a role here. We acknowledge this as a limitation, and have adjusted the wording of Discussion to state this more clearly (previously it mentioned “sensitivity bias”, which encompasses “social desirability bias”, though is a broader term):

“This study has a number of limitations, among them the possibility of social desirability bias affecting survey responses⁶⁹, though this concern is more pertinent for our issue polarization treatment results than our affective polarization findings, because only in the former case are our treatments similar in content to downstream attitude assessment²¹.”

I hope these comments help the researchers improve the paper

We thank reviewer 3. Indeed, the comments have been very helpful.

REVIEWERS' COMMENTS

Reviewer #1 (Remarks to the Author):

I enjoyed reading this revised manuscript examining polarization/misperception correction interventions in Brazil. I am very stratified with the authors' responses to my original comments and concerns. The inclusion of the information and clarifications that I asked for is good, and I find their defense of the use of fixed-effects panel modeling convincing. Moreover, I was happy to see the foregrounding of what I (still) believe is the primary contribution of this work: a gold standard model for conducting polarization and misperception reduction interventions outside of binary political-identity contexts like the United States, which dominates research and theorizing in this literature. I don't use the term gold standard here lightly. This paper is not merely generalizing past findings or slightly modifying existing theory and method, it's presenting a carefully thought out and integrated theoretical and methodological framework for generalizing (and challenging) a literature that is, in my opinion, deeply hampered by its US-centrism.

I find the paper well written, the research exceptionally thorough and expertly conducted, and the potential for significant contributions high.

Response: We thank Reviewer 1 for these comments, which have inspired a change in the manuscript title to better draw readers' attention to our primary contribution. The title now reads: "Sociopolitical polarization around elections and correction of controversial-policy misperceptions: considering a multi-definitional political-group context, Brazil".

Reviewer #2 (Remarks to the Author):

The authors addressed all my previous comments in a satisfactory manner. I'm still not convinced that the results of the manuscript are novel enough to warrant publication in Nature Communications but I leave it to the editor to decide on that matter.

Response: We are glad to have satisfied Review 2's requests.

Reviewer #3 (Remarks to the Author):

I appreciate the work the authors put into revising the manuscript. All my recommendations were thoroughly responded, and I am satisfied with the authors' responses. I believe this paper meets the requirements for publication at Nature Communications

My only remaining suggestion is for the authors to incorporate the results using the

alternative specifications for misinformation intervention in the supplemental materials. This is an interesting robustness finding, and readers should have access to it.

Thank you for the opportunity to revise this manuscript.

Response: We are glad to have satisfied Review 3's requests. We thank Reviewer 3 for the suggestion, and note that results using alternative specifications for misinformation intervention presented in the supplementary materials, (specifically Supplementary Fig. 16), are in fact referred to in the main manuscript, in the sentence, "Results remain largely unchanged using mixed-effects models with random intercepts and trends, as recommended by Curran and Bauer (Supplementary Fig. 16)."

Reviewer #4 (Remarks to the Author):

Preface: I have conducted this review with a very narrow focus. I have exclusively considered the soundness of the Bayes factor and equivalence test procedures reported. I have also considered 2s remark:

"A smaller technical issue for me was the equivalence tests the authors used. I'd recommend using Bayes factors to test for absence of effects as these don't rely on relatively arbitrary decisions about the smallest effect size of interest."

Finally, I have a remark on the response to the question by reviewer 1:

"Is there a formal test of the (equivalency) claim that the interventions were similar in size to the post-election drop? This is something claimed in the title of the paper, but if there's a formal test of the equivalence in effect size, I missed it."

I defer all other considerations of research quality to the editor and the other reviewers.

Review remarks:

I would like to begin by stating a slight dissatisfaction with Reviewer 2s remark. The reviewer makes it sound like the switch from equivalence tests to Bayes factors is an objective technical correction. It is not. Whether to use Bayesian or frequentist procedures is a matter of epistemological stance, and BFs are in no way strictly superior to TOST for evaluating potential null effects. Bayes factors, like equivalence tests, rely on assumptions that can sometimes be rather arbitrary (e.g. the use of default priors, and what exactly constitutes "strong evidence"). In addition, the common BF approach adopted here gives up on one of the main goals of TOST equivalence testing – to evaluate the practical significance of results. Essentially, the BF herein reported is a Bayesian version of standard NHST. It can only tell you something about how likely it is that the effect is "zero" or "not zero". It is not clear whether a large BF implies a meaningful effect, or if a small BF implies a non-meaningful effect, since the BF is not tied to the SESOI. If you think the SESOI is weak anyway then perhaps this move is appropriate. However, I note that the SESOI was actually based on prior relevant

research in this paper. If the SESOI really was considered informative by the authors, it is too bad that it does not seem to be informing the Bayesian analyses at all.

That said, I do not wish to debate whether to use BFs or equivalence tests in the middle of a review process. Both approaches have their strengths and limitations. Speaking as someone who is generally familiar with the Bayesian rationale and BF computation, the procedure implemented here seems completely fine as far as I can see. If the authors want to rely on Bayesian evidence rather than the previously reported frequentist equivalence tests, I don't think it is a very big issue – especially if the SESOI was not very strongly justified. In any case, BFs and TOST results are reported together and along with visualizations of confidence intervals, which gives the reader ample opportunity to decide for themselves which results to rely more heavily on. I think this is a completely fine way to approach the issue. However, I note two problems with the analyses that I think should be addressed:

1. (Minor) The calculation procedure for the BFs should be reported in the main manuscript, not just in the supplementary table text.
2. It seems that BFs are only computed for the non-significant results. This is a common, but inappropriate way to proceed. All results of interest, significant and non-significant, should be subjected to the same BF/equivalence test analyses, and BFs for all results should be reported in the main manuscript. The reason all effects should be evaluated is that...
 - a. (if testing for equivalence with SESOI) there could be results that are statistically significant from zero, but not significantly different from SESOI.
 - b. (if relying on Bayesian evidence) there could be there could be results that are statistically significant from zero, but for which there is not strong Bayesian evidence that the effect is different from zero.

Response: We thank Reviewer 4 for these very helpful points. First of all, we now provide BFs and equivalence test results for all results in Supplementary Table 5, to transparently address these issues as fully as possible. Due to length considerations, in the main manuscript, we report BFs for all null findings, and any instance where classical statistical results are not supported by BFs, which we also draw attention in the prose. This is now explained with a small addition to paragraph 4 of the Results section:

“We used linear regression models with standard errors clustered by respondent to test our hypotheses, and we conducted Bayesian analyses to provide complementary insights. In this section we present classical test statistics and p-values for all results. To conserve space in the main manuscript, we include Bayes factors for every null classical test result, and for classical test results that are significant but not supported by the Bayes factors (for Bayes factors pertaining to other significant findings and to equivalence tests, see Supplementary Information).”

We have also provide information in Methods to more fully explain our approach.

Regarding the question by reviewer 1, the authors reply “Bayes Factor analyses provide moderate to (mostly) strong evidence in favour of the null, as noted in Supplementary Table 5”. I don’t see any BFs in Supplementary table 5 that directly corresponds to an effect size comparison between treatment- and time effects. If I have just missed it, feel free to correct me. If not, some formal quantitative assessment of effect similarity between treatment- and time effects should be added to the analyses.

Finally, I find it a bit strange that the analyses and visualizations that directly bear on the title of the paper are relegated to supplementary materials. Unless there is some specific reason for this, I would recommend moving these results into the main manuscript.

I hope the authors find these comments helpful.

Signed: Peder M. Isager (I always sign my reviews)

Response: Thank you for drawing attention to this. After much discussion, and guidance from the editor, we decided to change the title of the manuscript because the review process had helpfully highlighted a mismatch of intention between the title and the main purpose of the manuscript.

We originally chose the word “similar” in the initial title carefully, for its looseness in bringing together the two parts of the paper: measurement of dynamic change, and treatment effects. However, comparison is not a central purpose of the paper.

At the time, we chose the word “similar” carefully, rather than selecting a word that would imply more specific claims of sameness or of difference. With just one word, we were referring to results pertaining to 4 different treatment conditions compared to dynamic change - and we could not think of another accurate single, summary word to describe the variation in these comparisons.

Having changed the title to remove the idea of comparison between the treatment effects and dynamic change, we do not think it worth including analyses nor visualisations of the comparisons. Hence, Supplementary Figure 17 (which illustrated the comparison, also providing equivalence tests) has now been removed.